# Maternal antibiotic exposure enhances ILC2 activation in neonates via downregulation of IFN1 signaling

Haixu Xu[1,6], Xianfu Yi[2,6], Zhaohai Cui[1,6], Hui Li[1], Lin Zhu[1], Lijuan Zhang[1], JiaLe Chen[1], Xutong Fan[2], Pan Zhou[1], Mulin Jun Li[2], Ying Yu[3], Qiang Liu[4], Dandan Huang[3] ✉, Zhi Yao[1] ✉ & Jie Zhou[1,5] ✉

Microbiota have an important function in shaping and priming neonatal immunity, although the cellular and molecular mechanisms underlying these effects remain obscure. Here we report that prenatal antibiotic exposure causes significant elevation of group 2 innate lymphoid cells (ILC2s) in neonatal lungs, in both cell numbers and functionality. Downregulation of type 1 interferon signaling in ILC2s due to diminished production of microbiota-derived butyrate represents the underlying mechanism. Mice lacking butyrate receptor GPR41 (*Gpr41*[-/-]) or type 1 interferon receptor IFNAR1 (*Ifnar1*[-/-]) recapitulate the phenotype of neonatal ILC2s upon maternal antibiotic exposure. Furthermore, prenatal antibiotic exposure induces epigenetic changes in ILC2s and has a long-lasting deteriorative effect on allergic airway inflammation in adult offspring. Prenatal supplementation of butyrate ameliorates airway inflammation in adult mice born to antibiotic-exposed dams. These observations demonstrate an essential role for the microbiota in the control of type 2 innate immunity at the neonatal stage, which suggests a therapeutic window for treating asthma in early life.

The maternal microbiota play an essential role in the instruction of immune system development in early life. Both clinical and experimental evidence has demonstrated that disruption of maternal microbiota due to antibiotic exposure during pregnancy imposes an increased risk to immune-related disorders in adult offspring, such as allergic airway inflammation[1]. However, the impact of maternal microbiota on neonatal immunity, as well as its contribution to disease susceptibility in adulthood, remain to be fully elucidated.

It has been demonstrated that neonates are biased towards type 2 immunity, which explains their higher susceptibility to allergic inflammation in childhood[2,3]. The enhanced type 2 immunity is partially associated with maternal-fetal immune adaptation due to hormonal changes during pregnancy[4]. Group 2 innate lymphoid cells (ILC2s) play an important role in the initiation and progression of allergic inflammation by secreting type 2 effector cytokines[5]. The activation of ILC2s is driven by danger signaling molecules such as interleukin (IL)−33, IL-25, and thymic stromal lymphopoietin, which does not involve antigen presentation and is therefore rapid[6–8]. Kleer et al. reported that the numbers of ILC2s peaked during the alveolar phase of lung development at postnatal day 3 to day 14 in mice, and

[1]Department of Immunology, Tianjin Institute of Immunology, Key Laboratory of Immune Microenvironment and Disease of the Ministry of Education, State Key Laboratory of Experimental Hematology, School of Basic Medical Sciences, Tianjin Medical University, Tianjin 300070, China. [2]Department of Bioinformatics, School of Basic Medical Sciences, Tianjin Medical University, Tianjin 300070, China. [3]Department of Pharmacology, School of Basic Medical Sciences, Tianjin Medical University, Tianjin 300070, China. [4]Department of Neurology, Institute of Neuroimmunology, Tianjin Medical University General Hospital, Tianjin 300052, China. [5]Department of Neonatology, Guangzhou Key Laboratory of Neonatal Intestinal Diseases, the Third Affiliated Hospital of Guangzhou Medical University, Guangzhou 510150, China. [6]These authors contributed equally: Haixu Xu, Xianfu Yi, Zhaohai Cui. ✉e-mail: mikey.huang2011@gmail.com; yaozhi@tmu.edu.cn; zhoujie@tmu.edu.cn

then declined after weaning[9]. The neonatal lung ILC2s were activated by IL-33 and displayed higher functionality than adult ILC2s[9,10]. Activation of neonatal ILC2s by IL-33 produced long-lasting effect on allergic inflammation in adults[11]. The dynamic regulation of neonatal ILC2s remains to be fully understood.

Based on the facts that the microbiota plays a critical role in the programming of the immune system in early life[12], and that expansion of ILC2s with enhanced activation occurs during this crucial window, we asked whether neonatal ILC2s are subject to regulation by microbiota. In this work, antibiotic exposure during pregnancy profoundly enhances ILC2 responses in neonates, which results in long-term effects and subsequently translates into higher susceptibility to allergic airway inflammation in adult offspring. The attenuated production of microbiota-derived butyrate and consequently downregulation of type 1 interferon (IFN1) signaling in neonatal ILC2s represents the underlying mechanism. Supplementation with butyrate during the perinatal stage prevents the allergic inflammation in adulthood. These observations provide insights into the importance of the microbiota in the suppression of type 2 innate immunity in early life.

## Results

### Prenatal antibiotic exposure enhances ILC2 responses in neonates

To explore whether prenatal antibiotic exposure affects type 2 innate immunity in offspring, pregnant mice were exposed to antibiotic cocktail (Abx) in the drinking water from embryonic day 10 to day 14; Abx was then removed until birth (Fig. 1a). The antibiotic cocktail included ampicillin (1 g/L), metronidazole (1 g/L), neomycin (1 g/L), and vancomycin (0.5 g/L)[13,14], and its efficiency of eliminating the gut commensal microflora was confirmed by qRT-PCR analysis of 16S ribosomal RNA (16S rRNA) (Supplementary Fig. 1a). Pups born to Abx dams exhibited no noticeable difference in body weight compared with control littermates (Supplementary Fig. 1b), indicating that maternal Abx we used did not impact the general embryonic growth. Flow cytometric analysis revealed that the proportions and absolute numbers of ILC2s (CD45$^+$Lin$^-$CD90.2$^+$CD25$^+$GATA3$^+$) in lungs changed dynamically after birth, peaking at postnatal day 14 (Supplementary Fig. 1c), which was consistent with a previous report[10]. Pups born to Abx dams displayed significantly higher numbers of ILC2s from postnatal day 3 to day 21 (Supplementary Fig. 1c). The production of effector cytokines and proliferation of lung ILC2s were clearly enhanced in pups born to Abx dams, which consequently led to increased infiltration of eosinophils into lungs (Supplementary Fig. 1d–f). For further confirmation, feces from healthy control dams were transplanted into Abx-treated dams (Fig. 1a). The success of fecal transplantation was confirmed by evaluation of the total microbiota (Supplementary Fig. 1a). Results showed that the reconstitution of gut microbiota in Abx-treated dams profoundly reduced the abundance of lung ILC2s in neonates (Fig. 1b), and the proliferation and effector cytokine production of lung ILC2s in pups were consistently decreased upon fecal transplantation (Fig. 1c). In line with the diminished ILC2 responses, the magnitude of allergic inflammation in lungs was clearly relieved by fecal transplantation, as evidenced by reduced eosinophilic infiltration and lower type 2 cytokine concentration in lung homogenates (Fig. 1d, e). The remission of tissue inflammation was further confirmed by hematoxylin-eosin (H&E) staining of lung sections (Fig. 1f). In addition, the elevation of neonatal ILC2s by maternal antibiotic exposure was also observed in small intestine, but not in adipose tissue (Supplementary Fig. 1g). Besides, there were minimal differences in the absolute numbers of ILC1s and ILC3s in intestines between pups from Abx and control dams (Supplementary Fig. 1h). Other types of immune cells in lungs also failed to show noticeable differences between Abx and control groups (Supplementary Fig. 1i). Moreover, no significant differences were observed in ILC2 progenitors in bone marrow (Supplementary Fig. 1j). These observations indicate that

maternal antibiotic exposure enhances ILC2 responses at mucosal barriers in neonates.

The effect of the microbiota on neonatal ILC2 responses was further confirmed using germ-free (GF) mice. Pups born to GF mice displayed significantly higher numbers of total ILC2s and activated ILC2s (IL-13$^+$IL-5$^+$ILC2), as well as infiltrated eosinophils in lungs, as compared with specific-pathogen-free (SPF) mice and co-housed GF littermates under steady state (Fig. 1g–i). The aggravated inflammatory responses in GF mice were confirmed by H&E staining and histological scoring of lungs (Fig. 1j).

### ILC2s from neonates born to Abx mothers display enhanced functionality

To further determine the effect of prenatal Abx exposure on neonatal ILC2 responses, mice with ILC2 deficiency (Rora$^{fl/fl}$ Il7r$^{Cre}$ mice) were used (Fig. 2a). We found that Rora$^{fl/fl}$ Il7r$^{Cre}$ neonates, not Rora$^{fl/fl}$ littermates control, were resistant to prenatal antibiotic-induced pulmonary inflammation, as represented by the comparable levels of eosinophils and concentrations of type 2 cytokines in lung homogenates, as well as histological staining of lungs (Fig. 2b–d). These results demonstrate that neonatal ILC2s contribute to the higher airway inflammation following prenatal antibiotic exposure.

For further confirmation, the same numbers of lung ILC2s from pups born to Abx dams and control dams were adoptively transferred into the immunodeficient non-obese diabetic (NOD)-Prkdc$^{em26Cd52}$Il2rg$^{em26Cd22}$/Nju (NCG) mice lacking lymphocytes, followed by intranasal administration of IL-33 or PBS vehicle for 3 consecutive days (Fig. 2e). The absolute cell counts of lung ILC2s in the recipients were significantly higher in IL-33-treated groups than PBS control, regardless of whether cells were from Abx or control groups (Fig. 2f). However, ILC2s from Abx dams displayed enhanced responses to IL-33 challenge compared with those from PBS control, as indicated by the elevated ILC2 numbers (Fig. 2f) and subsequent higher type 2 inflammation (Fig. 2g, h). Histological staining of lungs further confirmed the aggravated pulmonary inflammation in recipients of ILC2s from Abx group (Fig. 2i). These results support the idea that neonatal ILC2s born to Abx dams display enhanced functionality.

### Breast milk derived factors contribute to the effects of microbiota on neonatal ILC2s

Breast milk contains microbiota-derived metabolites, which impact the development of infant immunity[15]. We next investigated whether the effect of the microbiota on neonatal ILC2s was prenatally or postnatally determined. Cross-fostering experiments were performed: pups born to Abx dams or controls were switched at birth (Fig. 3a). Results revealed that pups born to control dams but cross-fostered to Abx dams (A → C) exhibited dramatic elevation of total ILC2s and activated ILC2s in lungs compared with those reared by control dams (C → C) (Fig. 3b, c). In contrast, cross-fostering pups born to Abx dams with control dams (C → A) lowered the numbers of lung ILC2s and dampened their responses, as compared with littermates reared by Abx dams (A → A) (Fig. 3b, c). The changes of ILC2 responses subsequently translated into the severity of airway inflammatory parameters (Fig. 3d–f). These results indicate that the phenotypes of ILC2s and allergic inflammation in neonates were largely determined by treatments given to foster mothers, raising the possibility that certain breast milk derived factors may determine ILC2 responses and pulmonary homeostasis in neonatal offspring.

### Maternal antibiotic exposure downregulates IFN1 signaling in neonatal ILC2s

To explore the mechanisms underlying the effect of microbiota on neonatal ILC2s, lung ILC2s from neonates born to Abx dams and those born to control dams were subjected to transcriptional profiling by SMART RNA sequencing (SMART-seq) (Supplementary Fig. 2a).

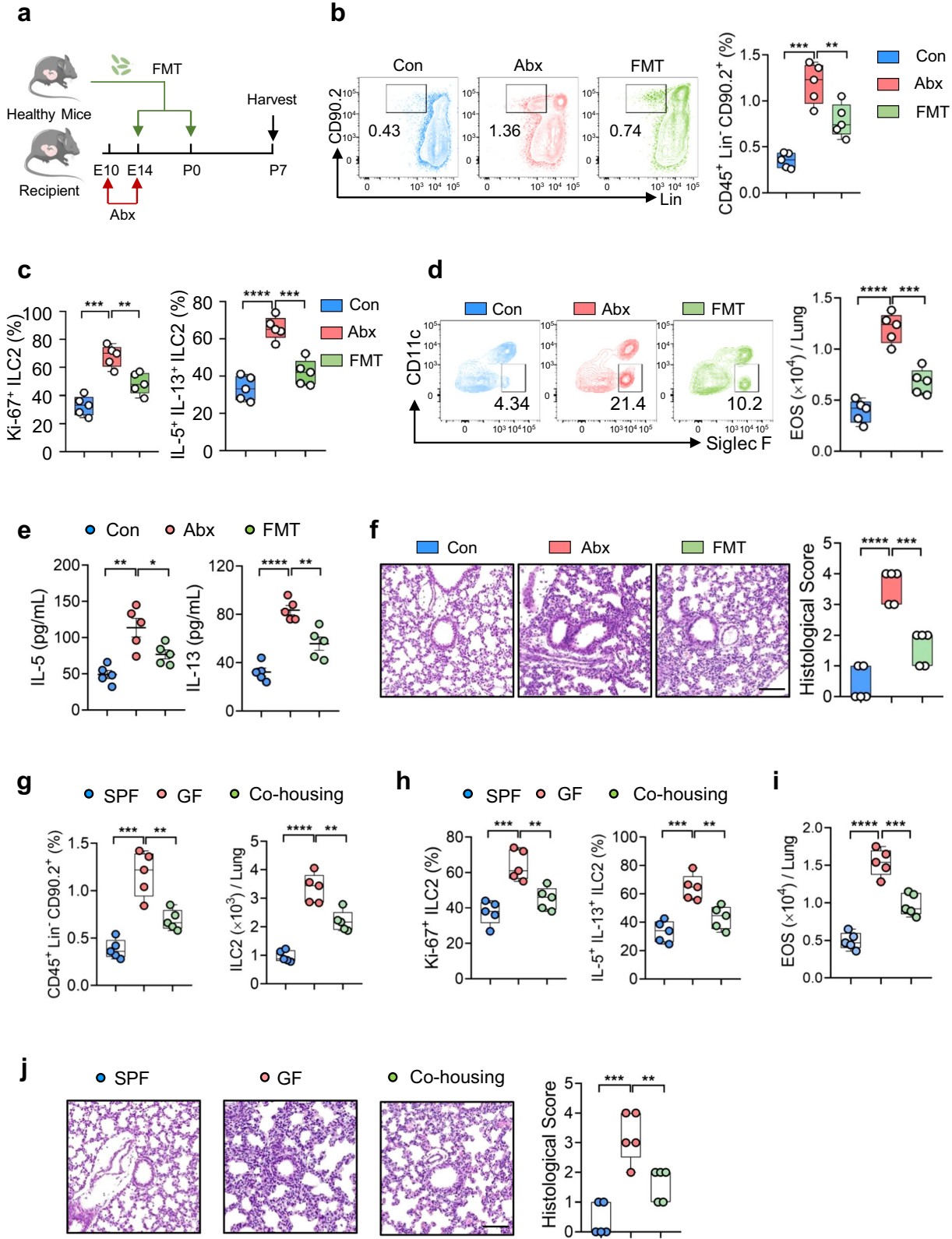

Volcano plot analyses revealed that known effector genes of type 2 immunity such as *Il5*, *Il13*, *Il9*, and *Tcf7* were dramatically upregulated in ILC2s from Abx group (Fig. 4a), in agreement with the enhanced ILC2 responses observed (Fig. 1). Interestingly, some important regulators of IFN1 signaling were dramatically downregulated in neonatal ILC2s from Abx mothers, including *Ifnb1*, *Ifnar1*, *Stat2*, *Stat1*, *Ifit2*, *Ifit1*, *Mx1*, *Isg15*, *Isg20*, *Irf3*, and *Irf7* (Fig. 4a,

Supplementary Fig. 2b). Kyoto Encyclopedia of Genes and Genomes (KEGG) analyses and gene set enrichment analysis (GSEA) further indicated the downregulation of IFN1 signaling in ILC2s from pups born to Abx dams (Fig. 4b, c). In agreement with their higher proliferative status, cell cycle pathway was upregulated in ILC2s from Abx group (Fig. 4b). It was reported that there is direct interaction between ILC2s and CD4+ T cells, activated ILC2s could promote T cell responses

**Fig. 1 | Prenatal antibiotic exposure enhances ILC2 responses in neonates.**
**a** Illustration of experimental model: pregnant mice (n = 5) were given an antibiotic cocktail (Abx) during day 10 to 14 of gestation stage. Fecal contents pooled from control dams were transplanted into Abx-treated pregnant dams by a single oral gavage until delivery (E14 - P0), and pups were sacrificed at postnatal day 7 (P7). The phenotype of ILC2s in lungs and allergic airway inflammation were evaluated (**b**–**f**). **b** Flow cytometric analysis of lung ILC2s (CD45+Lin-CD90.2+CD25+GATA3+) from the neonates. Lin-CD90.2+ gating is shown. **c** Flow cytometric analysis of the proliferation (left) and effector cytokine production (right) of ILC2s. **d** Frequencies and absolute numbers of eosinophils (CD45+CD11c-Siglec F+) in lungs were analyzed by flow cytometry. **e** The amounts of IL-5 and IL-13 in lung homogenates were measured by ELISA. **f** H&E staining of lung tissues (bar, 100 μm), and the histological scoring. **g**–**j** Wild-type SPF male mice were co-housed with GF females, and their offspring were sacrificed at P7. The phenotype of ILC2s and allergic inflammation in

lungs were evaluated. **g** Flow cytometric analysis of the proportions and absolute numbers of lung ILC2s. **h** Flow cytometric analysis of the frequencies of proliferative ILC2s (Ki-67+) and cytokine-producing ILC2s (IL-5+ IL-13+) in lungs. **i** Absolute numbers of eosinophil (CD45+CD11c-Siglec F+) in lungs were analyzed by flow cytometry. **j** H&E staining of lung tissues (bar, 100 μm) and the histological score. In all panels, 2–3 independent experiments were performed. In Fig. 1e, the data are presented as the mean ± SEM values, by unpaired two-tailed Student's *t* test. For box plots, the data are shown as "Min to Max, show all points". For box plots, the midline represents the median; box represents the interquartile range (IQR) between the first and third quartiles, and whiskers represent the lowest or highest values within 1.5 times IQR from the first or third quartiles (**b**, **c**, **d**, **f**, **g**, **h**, **i**, **j**). **P < 0.01; ***P < 0.001; ****P < 0.0001 by unpaired two-tailed Student's *t* test. Statistical source data are provided in Source Data.

via a variety of mechanisms[16]. T cell receptor signaling was also upregulated in ILC2s from Abx group (Fig. 4b). The dampened IFN1 signaling was confirmed by qRT-PCR (Supplementary Fig. 2c). ELISA assay showed that the amounts IFN-β in the culture supernatants were lower in neonatal ILC2s from Abx group, as compared with control (Fig. 4d). In addition, neonatal lung ILC2s displayed upregulation of *Ifnb1* expression and IFN-β production under HSV infection, which was comparable with splenic macrophages (Supplementary Fig. 2d, e). The levels of STAT1/STAT2 phosphorylation, key intracellular events upstream of IFN1 signaling[17], was decreased in neonatal lung ILC2s after prenatal Abx exposure (Supplementary Fig. 2f). Furthermore, administration of IFN-β in vitro clearly suppressed the production of IL-5 and IL-13 from cultured neonatal lung ILC2s in dose-dependent manner (Fig. 4e). Consistent with the in vitro observations, administration of IFN-β efficiently reduced ILC2 frequencies, lowered their responses, and consequently alleviated airway inflammation in WT pups born from Abx dams (Fig. 4f–i, Supplementary Fig. 2g). Neonatal mice with IFN1 receptor IFNAR1 deletion (*Ifnar1*-/- mice) displayed elevated ILC2 responses and aggravated type 2 inflammation in lungs under steady state; furthermore, they were resistant to maternal Abx exposure and postnatal IFN-β supplementation (Fig. 4f–i). These observations indicate that IFN-β, via its receptor IFNAR1, may mediate the effect of the microbiota on ILC2s in neonates.

## Butyrate suppresses neonatal ILC2 responses via upregulation of IFN1 signaling

Nutrients, including microbiota-derived metabolites, in breast milk play a critical role in shaping the development of the immune system in early life, therefore impacting the susceptibility of immune disorders in offspring, which is the basic concept of the window of opportunity in the field of infant immunity[18]. To further delineate which maternal factor dictates ILC2 responses in the offspring, serum from Abx-treated dams and control dams was collected and evaluated by non-targeted metabolomic analysis. Volcano plots showed that short-chain fatty acids (SCFAs), including butyrate and its substrate lactate, propionate, and acetate, were profoundly downregulated in Abx-derived serum (Fig. 5a). On the other hand, the levels of leukotriene B4, an important inflammatory mediator derived from arachidonic acid metabolism, was upregulated in ILC2s from Abx group (Fig. 5a).

Since SCFAs are derived from fermentation of dietary fiber by the gut microbiota, and have therapeutic value in the prevention of airway inflammation[19], we next evaluated changes of bacterial composition in dams with or without antibiotic exposure. 16S rRNA sequencing showed that the composition and abundance of the microbiota were significantly different between Abx-treated dams and control dams, although α diversity did not show noticeable changes (Supplementary Fig. 3a, b). Furthermore, Abx-treated dams were characterized by a lower abundance of butyrate-producing *Firmicutes* phylum and specifically *Lactobacillaceae* family (Supplementary Fig. 3c, d). Consistent observations

were obtained in pups born to Abx dams, although the composition of microbiota was different between dams and pups (Supplementary Fig. 3e, f)[20]. These observations indicate that maternal antibiotic exposure diminished the production of microbiota-derived SCFAs.

Further targeted metabolomics analysis by LC-MS showed that butyrate was the markedly changed SCFAs in both serum (Fig. 5b) and breast milk (Fig. 5b) between Abx-treated dams and the control dams. Neonatal mice born to Abx dams also displayed significant lower levels of butyrate in the lungs as compared with control littermates (Supplementary Fig. 4a). These observations raise the possibility that the impaired production of butyrate due to prenatal antibiotic exposure contributes to the enhanced ILC2 responses in neonates. In vitro culture showed that butyrate downregulated the mRNA expression of ILC2 signature genes such as *Gata3*, *Il5*, and *Il13* in neonatal ILC2s (Fig. 5c). Furthermore, administration of butyrate to GF neonatal mice clearly reduced the abundance of ILC2s in lungs, and decreased their proliferation and effector cytokines production (Fig. 5d). The diminished ILC2 responses consequently resulted in a clear reduction of eosinophil infiltration (Fig. 5d), and lowered type 2 cytokines in lungs (Supplementary Fig. 4b). These results demonstrate that butyrate inhibits ILC2 responses in neonates.

Considering the observations that both butyrate and IFN1 signaling are causally linked to ILC2 responses upon maternal antibiotic exposure, we next investigated their regulatory relationship. Phosphorylation of STAT1 and STAT2, key transcription factors elicited by IFN1 signaling, was significantly elevated in neonatal ILC2s upon butyrate administration (Fig. 5e). Meanwhile, butyrate dramatically induced the transcription of interferon-stimulated genes (ISGs) in neonatal ILC2s in vitro, including *Ifnar1*, *Irf7*, *Mx1*, *Isg15* and *Ifnb1* (Fig. 5f). Administration of fludarabine and hydrocortisone, selective inhibitors of STAT1 and STAT2 respectively[21,22], clearly counteracted the effects of butyrate on the effector cytokines production of ILC2s (Fig. 5g).

SCFAs exert their biological function either through binding with G protein-coupled receptors such as GPR41, GPR43, and GPR109A, or through inhibition of histone deacetylase 9 (HDAC9), to elicit intracellular signaling[23]. Gene profiling showed that the transcripts of *Gpr41* were most abundant in neonatal ILC2s compared with *Gpr43* and *Gpr109a*, as well as *Hdac9* (Supplementary Fig. 4c). The abundant expression of GPR41 in neonatal ILC2s was further confirmed by flow cytometry and immunofluorescence staining (Fig. 5h). AR420626, a selective agonist of GPR41[24], efficiently diminished neonatal ILC2 responses, whereas the GPR43 agonist 4-CMTB[25] displayed no noticeable effect (Supplementary Fig. 4d). In addition, coadministration of GPR41 antagonist polyhydroxy butyrate[26] significantly enhanced ILC2 responses as compared with butyrate alone (Supplementary Fig. 4e). Consistently, butyrate supplementation in vivo counteracted the effect of maternal antibiotic exposure on ILC2 responses (Fig. 5i) and IFN1 signaling (Fig. 5j) in WT pups, not in *Gpr41*-/- littermates. These observations indicate that butyrate/GPR41 axis induces IFN1 signaling in neonatal ILC2s.

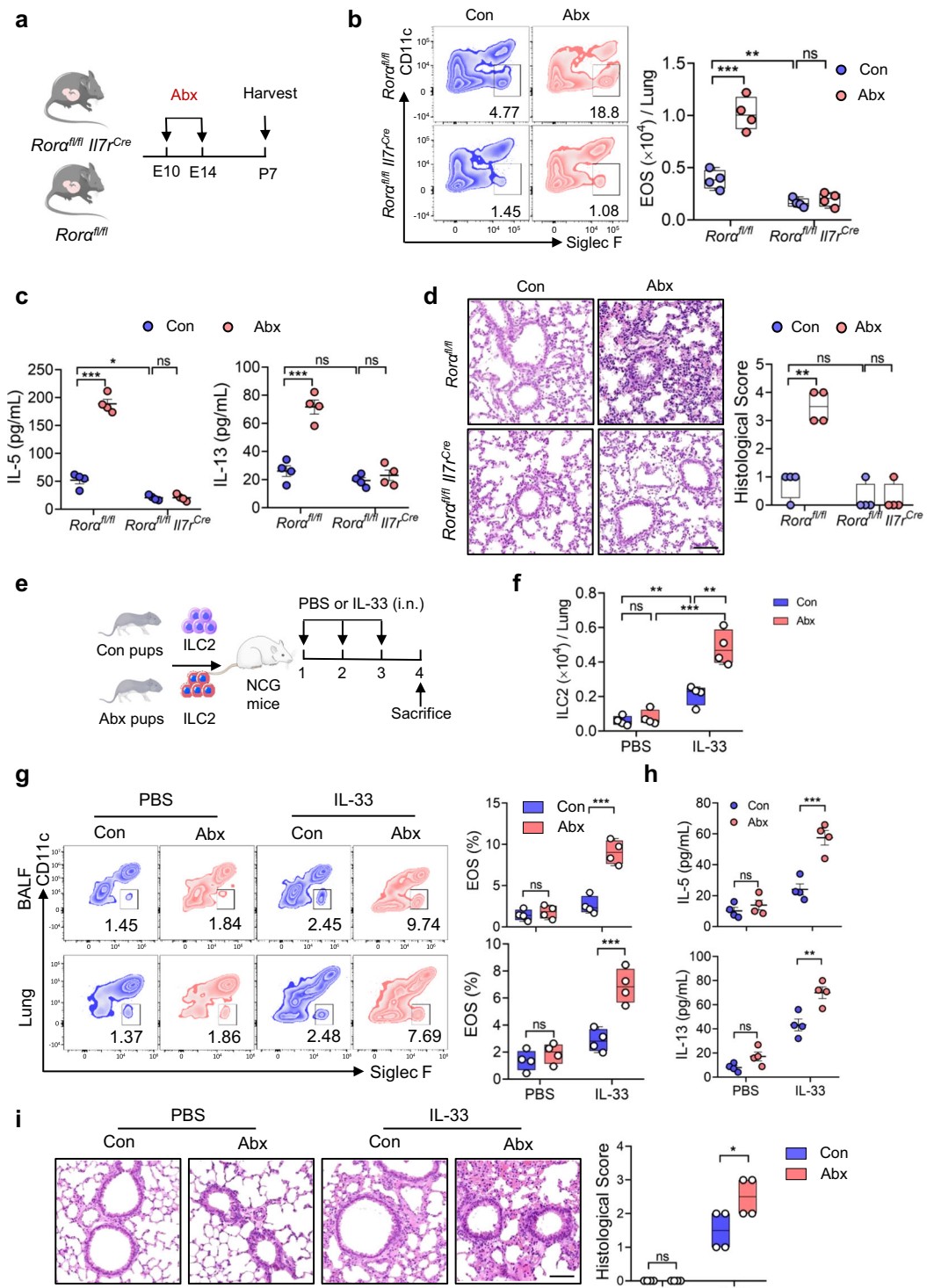

**Fig. 2 | ILC2s in neonates born to Abx mothers display enhanced functionality.**
**a** Schematic diagram of maternal antibiotic treatment of ILC2-deficient mice
(*Rora^fl/fl Il7r^Cre*) and *Rora^fl/fl* littermate controls (*n* = 4). Allergic airway inflammation,
including EOS numbers in lungs, amounts of type 2 cytokines in lungs and histo-
logical scoring of lungs were evaluated (**b**–**d**). **b** Frequencies and absolute numbers
of eosinophils in lungs were evaluated by flow cytometry. **c** Amounts of IL-5 and IL-
13 in lung homogenates were measured by ELISA. **d** H&E staining of lungs (bar,
100 μm) and histological scoring. **e** Illustration of experimental model: lung ILC2s
were sorted from neonatal mice from Abx-treated and control dams, followed by
transferred into NCG mice (1.5 × 10⁴ cells/mouse). NCG mice were then challenged
intranasally with IL-33 or PBS daily for 3 days (*n* = 4). The phenotype of ILC2 in lungs
and allergic airway inflammation were evaluated (**f**–**i**). **f** Flow cytometric analysis of

ILC2s in lungs. **g** Flow cytometric analysis of eosinophils in lungs and BALF.
**h** Amounts of IL-5 and IL-13 in BALF were determined by ELISA. **i** H&E staining of
lung tissues (bar, 100 μm) and the histological scoring. In Fig. 2c and h, the data are
presented as the mean ± SEM values, by unpaired two-tailed Student's *t* test. For
box plots, the data are shown as "Min to Max, show all points". For box plots, the
midline represents the median; box represents the interquartile range (IQR)
between the first and third quartiles, and whiskers represent the lowest or highest
values within 1.5 times IQR from the first or third quartiles (**b**, **d**, **f**, **g**, **i**).
*$P < 0.05$; **$P < 0.01$; ***$P < 0.001$ by unpaired two-tailed Student's *t* test or one-way
ANOVA followed by Tukey-Kramer multiple-comparisons test. Data are pre-
sentative of 2-3 independent experiments (**a**–**i**). Statistical source data are pro-
vided in Source Data.

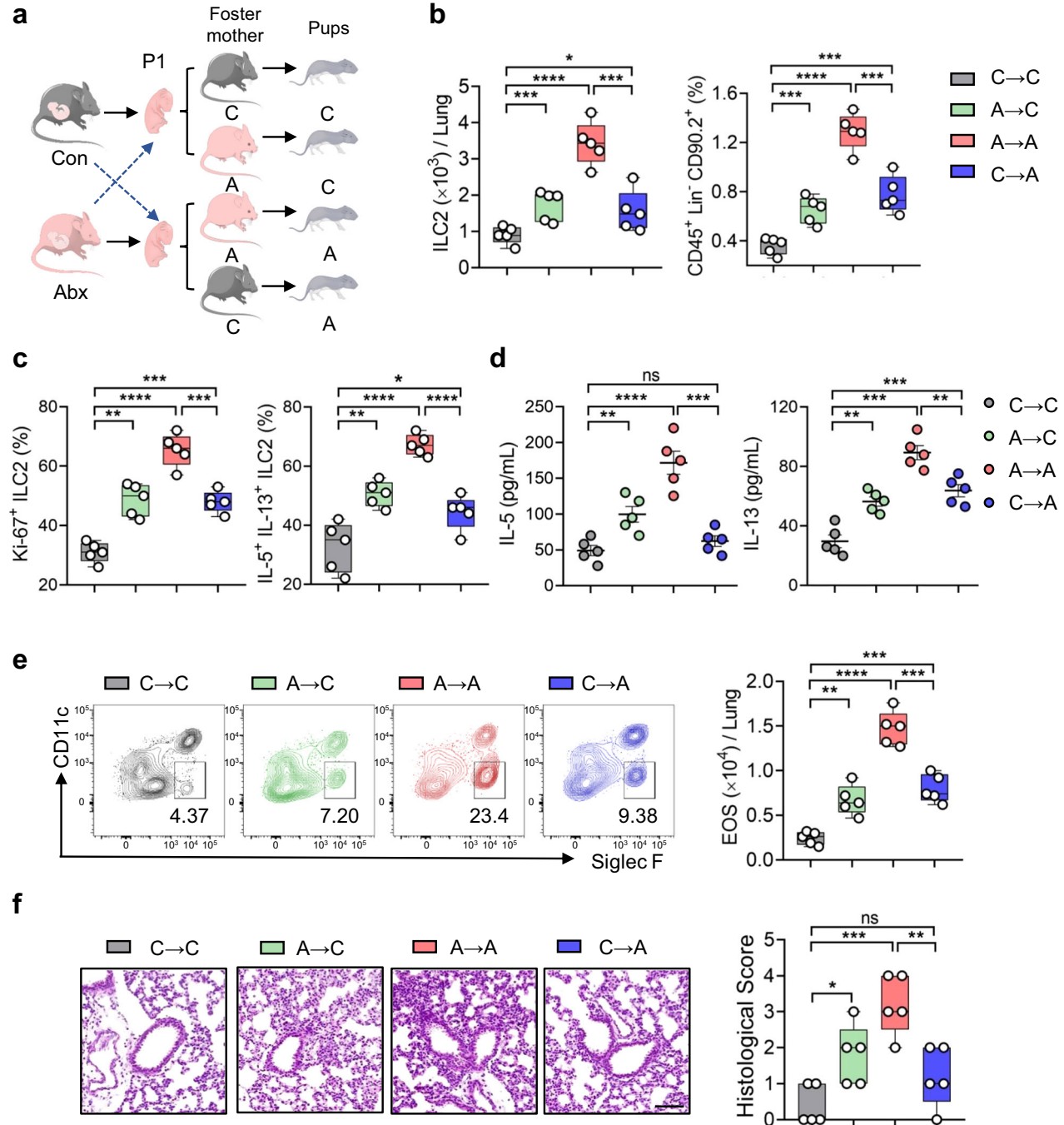

**Fig. 3 | Breast milk derived factors contribute to the effects of microbiota on neonatal ILC2s. a** Schematic diagram of cross-fostering experiments: pups born to antibiotic dams and control littermates were switched at postnatal day 1 (P1) (*n* = 5). The phenotype of ILC2 in lungs and allergic airway inflammation were evaluated (**b**–**f**). **b** Flow cytometric analysis of ILC2s in neonatal lungs. **c** Flow cytometric analysis of the frequency of proliferative ILC2s (Ki-67⁺) and cytokine-producing ILC2s (IL-5⁺IL-13⁺). **d** Amounts of IL-5 and IL-13 in lung homogenates were determined by ELISA. **e** Flow cytometric analysis of eosinophils in lungs. **f** H&E staining of lung tissues (bar,

100 µm) and the histological scoring. In Fig. 3d, the data are presented as the mean ± SEM values, by unpaired two-tailed Student's *t* test. For box plots, the data are shown as "Min to Max, show all points". For box plots, the midline represents the median; box represents the interquartile range (IQR) between the first and third quartiles, and whiskers represent the lowest or highest values within 1.5 times IQR from the first or third quartiles (**b**, **c**, **e**, **f**). *$P < 0.05$; **$P < 0.01$; ***$P < 0.001$; ****$P < 0.0001$ by unpaired two-tailed Student's *t* test. Data are presentative of 2-3 independent experiments (**a**–**f**). Statistical source data are provided in Source Data.

## Maternal antibiotic exposure induces epigenetic changes in ILC2s and exerts long-term effects on allergic inflammation in adult offspring

We next investigated whether maternal antibiotic exposure could induce long-lasting effects on ILC2 responses and pulmonary inflammation. Lung ILC2s from adult offspring born to Abx dams and control dams were subjected to ATAC sequencing (ATAC-seq) to evaluate their

chromatin accessibility (Fig. 6a). A total of 3431 and 7061 peaks were distinctly open in ILC2s from Abx and control groups respectively (Supplementary Fig. 5a). In addition, the distribution of open chromatin regions across the genome was largely unchanged (Supplementary Fig. 5b). The differentially open peaks located in the enhancer and promoter regions were further compared. ILC2s from Abx group showed enhanced chromatin accessibility in ILC2 hallmark genes such

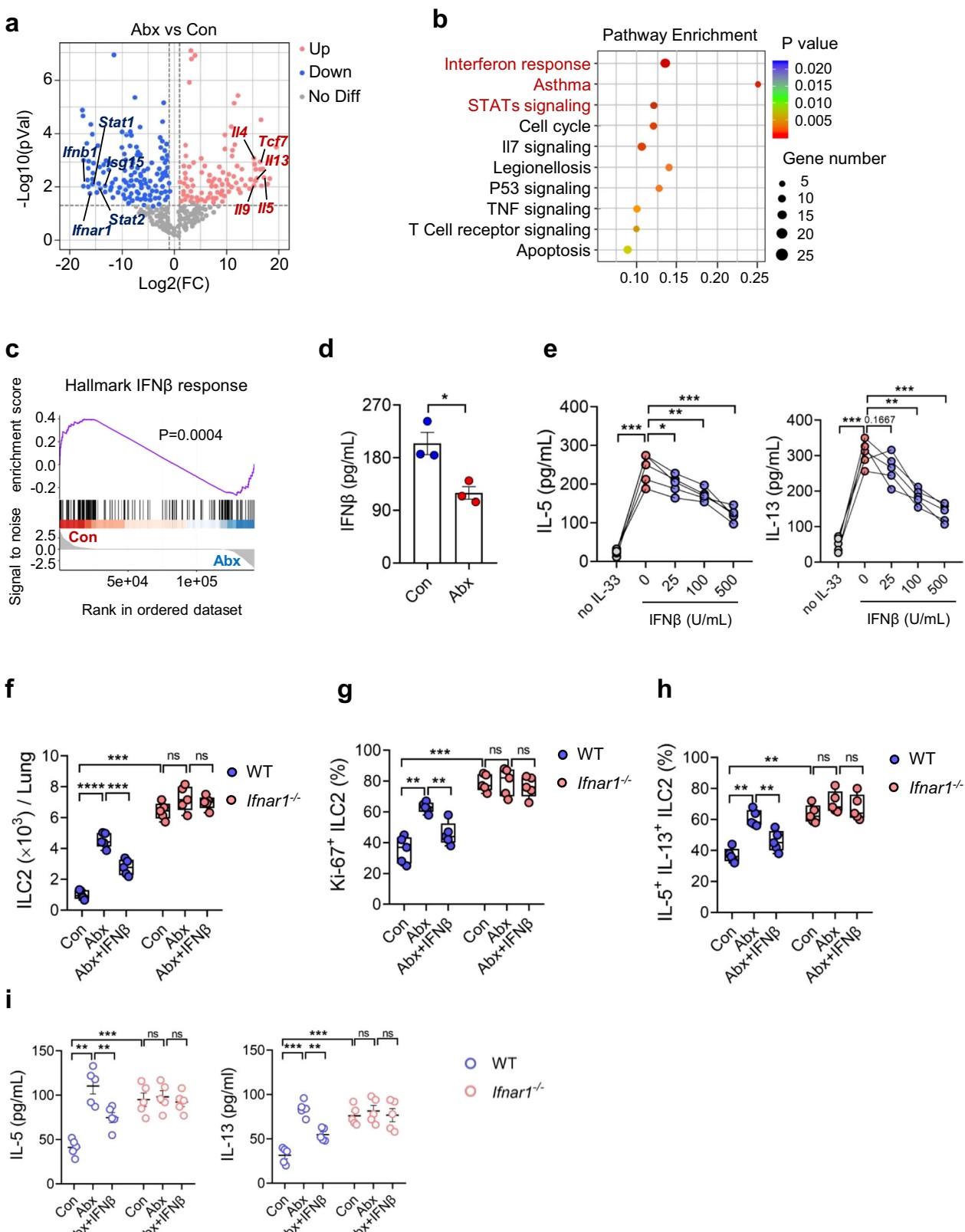

as *Il5* and *Icos*, whereas ILC2s from control group exhibited higher chromatin accessibility of IFN-stimulated genes (ISGs) such as *Mx1*, *Mx2* and *Oas2* (Supplementary Figs. 5c, 6b). These observations were consistent with mRNA transcriptional analysis by SMART-seq (Fig. 4). Moreover, Th2 signaling and ICOS signaling known to be closely related to ILC2 activation[5], were enriched in ILC2s from Abx group (Fig. 6c). We speculated the potential binding sites open to

transcriptional regulators that might contribute to these differences. HOMER de novo motif analysis identified multiple transcription factors important for ILC2 function, including *Gata3*, *Tcf7*, *Stat3*, *Stat5* and *Stat6*, *Gfilb*, *Nf-kb*, were obviously enriched in ILC2s from Abx group as compared with littermate control, the enrichment of *Isre* (Interferon-stimulated response elements) displayed reverse change (Fig. 6d, Supplementary Fig. 5d). These observations indicate that maternal

**Fig. 4 | Maternal antibiotic exposure downregulates IFN1 signaling in neonatal ILC2s. a–c** Lung ILC2s were sorted from neonates born to Abx or PBS dams, and transcriptional profiling was evaluated by SMART-seq. **a** Volcano plots presenting the differences between two groups. The negative binomial distribution was employed for the identification of differentially expressed genes through the use of DESeq2. The Benjamini-Hochberg method was applied for multiple testing correction. **b** Top 10 pathways enriched in KEGG analysis. The hypergeometric test was employed to assess the impact of differentially expressed genes on a pathway. The Benjamini-Hochberg method was applied for multiple testing correction. **c** Gene set enrichment analysis (GSEA) showing the downregulation of interferon β (IFNβ) signaling in Abx group. The one-sided Wilcoxon rank-sum test was employed for the enrichment analysis of the IFNβ signaling pathway within the framework of GSEA. **d** Amounts of IFNβ in the culture supernatants of lung ILC2s were measured by ELISA (*n* = 3). **e** The amounts of IL-5 and IL-13 in the culture supernatants of lung ILC2s in the presence of different dosages of IFNβ were measured by ELISA (*n* = 4).

**f–i** WT and *Ifnar1*[-/-] pregnant mice were subjected to Abx treatment, pups were intraperitoneally injected with IFNβ (1 × 10⁴ U/mouse) daily for 5 consecutive days. Pups were sacrificed at postnatal day 7 (*n* = 5). The absolute number of ILC2 (**f**), the frequency of Ki-67⁺ ILC2s (**g**), and the frequency of IL-5⁺ IL-13⁺ ILC2s (**h**) in lungs were analyzed by flow cytometry. **i** Amounts of IL-5 and IL-13 in BALF were determined by ELISA. In Fig. 4d, 4e and 4i, the data are presented as the mean ± SEM values, by unpaired two-tailed Student's *t* test. For box plots, the data are shown as "Min to Max, show all points". For box plots, the midline represents the median; box represents the interquartile range (IQR) between the first and third quartiles, and whiskers represent the lowest or highest values within 1.5 times IQR from the first or third quartiles (**f, g, h**). **P < 0.01; ***P < 0.001; ****P < 0.0001 by unpaired two-tailed Student's *t* test or one-way ANOVA followed by Tukey-Kramer multiple-comparisons test. Data are presentative of 2-3 independent experiments (**d–i**). Statistical source data are provided in Source Data.

antibiotic exposure induced epigenetic changes in ILC2s in adult offspring.

We next asked whether the induced epigenetic changes in offspring ILC2 could produce long-lasting effect on pulmonary inflammation in adult offspring, papain model[27] was employed to induce allergic airway inflammation (Fig. 6e). Flow cytometric analysis revealed that the abundance of lung ILC2s and their activation were significantly higher in adult mice born to Abx mothers as compared with control littermates, especially upon papain challenge (Fig. 6f, g). The enhanced ILC2 responses further aggravated the severity of airway inflammation in offspring born to Abx mothers (Fig. 6h–j).

Aim to evaluate the contribution of ILC2s to the aggravated airway inflammation caused by maternal antibiotic exposure, *Rora*^fl/fl^*Il7r*^Cre^ mice and *Rora*^fl/fl^ control littermates were used. In sharp contrast with the exacerbation of airway inflammation in *Rora*^fl/fl^ control littermates born to Abx dams, *Rora*^fl/fl^ *Il7r*^Cre^ pups born to Abx dams displayed much lower levels of pulmonary inflammation (Supplementary Fig. 5e–g). These observations suggest that maternal antibiotic exposure exerts a long-term protection in the control of airway inflammation in offspring, in which ILC2s play an important role.

**Therapeutic window of allergic airway inflammation in early life**

The impaired production of butyrate due to maternal antibiotic exposure prompted us to test whether butyrate supplementation has beneficial effect on the prevention of airway inflammation in adult offspring. For prenatal treatment, antibiotic-exposed dams were supplemented with butyrate (200 mM)[28] in drinking water from embryonic day 14 to postnatal day 7 (Prenatal group); alternatively, neonates were injected intraperitoneally with butyrate for 5 days (Neonatal group); For post-wean group, butyrate was administered in drinking water from postnatal day 21 for two weeks (Fig. 7a). Flow cytometric analysis showed that prenatal or neonatal administration of butyrate, rather than after weaning, significantly reduced the abundance of lung ILC2s and dampened their activation in offspring born to Abx-treated dams (Fig. 7b–d). As a result of the diminished ILC2 responses, the severity of airway inflammation was clearly relieved by prenatal or neonatal supplementation of butyrate (Fig. 7e, f). These observations indicate that SCFAs supplementation in early life could protect the offspring against allergic airway inflammation upon maternal dysbiosis.

## Discussion

Both clinical and animal experiments have demonstrated that the microbiota plays an important role in the prevention of allergic airway inflammation in offspring[29]. The cellular and molecular mechanisms of this phenomenon remain to be fully elucidated. In this study, we showed that the offspring mice born to maternal antibiotics exposure displayed higher susceptibility of allergic airway inflammation, in which enhanced ILC2 responses contributed to this phenotype.

Mechanistic studies showed that maternal antibiotic exposure reduced the production of SCFAs from mother, which downregulated IFN1 signaling in neonatal ILC2s via breast feeding. The downregulation of IFN1 signaling enhanced ILC2 responses and aggravated allergic airway inflammation in offspring. Perinatal supplementation of SCFAs counteracted the effects of maternal antibiotic exposure. These observations provide an early opportunity for the prevention of allergic inflammation.

The importance of maternal-derived SCFA in shaping the development of immune system in early life and therefore affecting the risk of immune-related disorders in offspring has been documented. SCFAs could regulate the function of dendritic cells and regulatory T cells, via the gut–lung axis, which contribute to an increased risk of allergic airway inflammation in adult offspring[19,29,30]. It remains to be explored whether the microbiota impacts innate immunity during infancy. Both the numbers and the function of ILC2s in infants were much higher compared with adulthood, which may contribute to the enhanced susceptibility of allergic asthma in children[9]. In this study, antibiotics exposure in the middle term of pregnancy profoundly elevated the numbers of ILC2s and enhanced ILC2 activation in neonates, which were functionally translated into aggravated airway inflammation. Mice with ILC2 deficiency were resistant to maternal antibiotic-induced type 2 inflammation in neonates. These observations highlighted the importance of neonatal ILC2s in the orchestration of pulmonary homeostasis in early life. The cross-talk between ILC2s with other types of immune cells during infancy, such as regulatory T cells and dendritic cells, deserves further investigation.

The microbiota colonization after birth was influenced by nutrients from breast milk and microbiota from mother and environment. The infant's microbiota and their metabolites affect their immune system development[31]. In this study, the elevated ILC2 responses in neonates due to maternal antibiotic exposure was not completely rescued after cross-fostering by control dams (compare C → A and C → C groups in Fig. 3), these results indicate that the contribution of prenatal factors to the observed ILC2 phenotypes in maternal antibiotic exposure model could not be excluded. Evaluation of microbiota from pups showed that SCFA-producing species were consistently reduced in those born from Abx dams, although the composition of microbiota from pups differed from dams as expected. Another possibility is that microbiota in the feces or skin of control dams could have been orally transferred to and affected the neonates.

Although we focused on the effects of microbiota on neonatal ILC2 in lungs, the potential role of microbiota in the regulation of other ILC subsets remains to be investigated. The role of ILC3 in respiratory diseases is less studied, they play an important role in tissue homeostasis, infection and inflammation in gut via secretion of type 3 cytokine IL-17 and IL-22[32,33]. In this study, we did not observe changes in the frequencies of ILC1 and ILC3 in intestines of neonates born of Abx and control dams. The potential effects of microbiota on the function

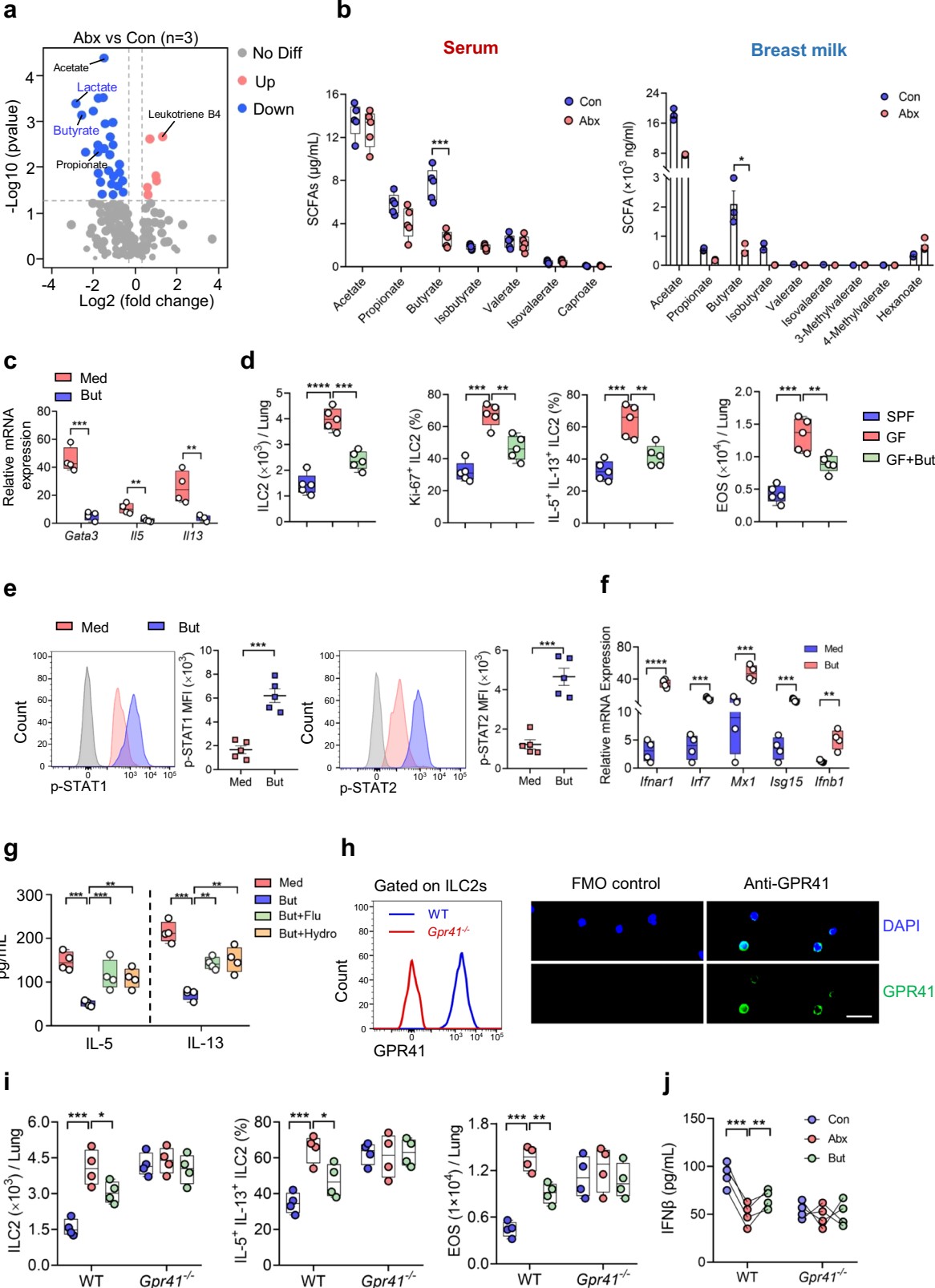

of other ILC subsets in offspring deserves further investigation, such as utilization of infectious model and colitis model. This will broaden our understanding about the comprehensive roles of microbiota in the development of innate immune system in early life.

It was reported that ILC2s expressed IFN1 receptor *Ifnar1*[34]. Previous studies have demonstrated that IFN1 signaling negatively regulated ILC2 responses and alleviated the pathogenesis of allergic

airway inflammation[34,35]. Administration of IFNα to neonatal mice prevented the development of allergic airway inflammation in IFNAR1 dependent manner[36]. However, whether IFN1 signaling regulates ILC2 in paracrine or autocrine manner, still remains to be investigated. In this study, IFN1 signaling in ILC2s from pups was downregulated in response to maternal antibiotic exposure, administration of IFNβ suppressed the production of effector

**Fig. 5 | Butyrate suppresses neonatal ILC2 responses via upregulation of IFN1 signaling. a,b** Metabolomics analysis was performed to evaluate the metabolites in breastmilk and serum from Abx-treated dams and control dams. **a** Volcano plot showing metabolites with differential abundance between Abx-treated dams and the controls (n = 3). The two-sided Wilcoxon rank-sum test was employed to identify differences. The Benjamini-Hochberg method was applied for multiple testing correction. **b** Targeted metabolomics analysis of SCFAs in serum (n = 5)/breast milk (n = 3) of Abx-treated dams and controls. **c** ILC2s from neonatal lungs were cultured in vitro with 100 ng/mL IL-33, 20 ng/mL IL-2 and 20 ng/mL IL-7 for 3 days, in the presence of butyrate (But, 2 mM) or medium control (Med). The mRNA expression of *Gata3*, *Il5*, and *Il13* was determined by qRT-PCR (n = 4). **d** GF newborn mice were intraperitoneally injected with butyrate (100 mg/kg) daily for 5 consecutive days, and pups were sacrificed at postnatal day 7 (n = 5). The absolute numbers of ILC2s, the frequency of Ki-67$^+$ ILC2s and IL-5$^+$IL-13$^+$ ILC2s, as well as eosinophils in lungs were evaluated by flow cytometry. **e–g** Lung ILC2s from neonatal mice were cultured with 100 ng/mL IL-33, 20 ng/mL IL-2, and 20 ng/mL IL-7 for 3 days in the presence or absence of butyrate (2 mM). **e** Phosphorylation of STAT1 and STAT2 was determined by flow cytometry (n = 5). **f** mRNA expression of *Ifnar1*, *Irf7*, *Mx1*, *Isg15* and *Ifnb1* was determined by qRT-PCR (n = 4). **g** Lung ILC2s from neo-

neonates were cultured with IL-2, IL-7, and IL-33 for 3 days in the presence or absence of butyrate (2 mM), fludarabine (Flu, 5 μM), or hydrocortisone (Hydro, 0.4 mg/mL) for 12 h. Amounts of IL-5 and IL-13 in supernatants were determined by ELISA (n = 4). **h** Expression of GPR41 on neonatal ILC2s was evaluated by flow cytometry (left) and immunofluorescence. Scale bar, 10 μm (right). **i–j** *Gpr41*$^{-/-}$ pregnant mice and WT control were subjected to antibiotic treatment; pups were intraperitoneally injected with butyrate (100 mg/kg) daily for 5 consecutive days and were sacrificed at PND7 (n = 4). Absolute numbers of ILC2s (**i**-left) and the frequency of IL-5$^+$IL-13$^+$ ILC2s (**i**-middle) and absolute numbers of eosinophils (**i**-right) were evaluated by flow cytometry. Amounts of IFNβ in lung homogenates were measured by ELISA (**j**). In Fig. 5e and 5j, the data are presented as the mean ± SEM values, by unpaired two-tailed Student's t test. For box plots, the data are shown as "Min to Max, show all points". For box plots, the midline represents the median; box represents the interquartile range (IQR) between the first and third quartiles, and whiskers represent the lowest or highest values within 1.5 times IQR from the first or third quartiles (**b, c, d, f, g, i**). *P < 0.05; **P < 0.01; ***P < 0.001; ****P < 0.0001 by unpaired two-tailed Student's t test (**d, g, i**) or Mann–Whitney U test (**b, c, f**). Data are presentative of 2-3 independent experiments (**c–j**). Statistical source data are provided in Source Data.

cytokines from ILC2 both in vitro and in vivo. The levels of IFNβ in the culture supernatants from neonatal ILC2s were comparable to those from macrophages in response to HSV infection. These results indicate that IFN1 signaling may regulate ILC2s via autocrine manner, although paracrine regulation is also possible.

Microbiota-derived SCFAs exert broad immunomodulatory functions to shape the immune system, which play a beneficial role in the hosts. Butyrate could drive colonic Treg differentiation and maintain gut homeostasis[37]. After absorption into enterocytes, butyrate enters the circulation and regulates immune responses in the peripheral tissues[38]. Butyrate could bind with its receptors expressed on target immune cells or via the inhibition of histone deacetylases (HDAC), elicits intracellular signaling and modulates the function of immune cells[38]. Butyrate produced by microbiota during pregnancy could secret into breast milk and pass to the neonates during feeding. Butyrate in human breast milk was considered to be protective biomarker for food allergy[39]. In addition to immune system, maternal SCFAs also play an important role in the health of offspring, such as neuroplasticity, cognitive and social functions[40,41]. In this study, we demonstrated the protective role of maternal butyrate in the allergic airway inflammation in offspring. Considering that SCFAs could be produced by microbiota under steady state condition, the effect of butyrate treatment on pregnant mice without antibiotic treatment was not evaluated, when butyrate is sufficient. In order to avoid the adverse effects of antibiotic exposure to early development of fetus and avoid fetus losses, antibiotic exposure was administered to pregnant mice from embryonic day 10 to day 14[13,14].

Overall, this study sheds further light on the intricate relationship between the microbiota and type 2 innate immunity during infancy. It uncovers an important cellular and molecular mechanism underlying the causal relationship between maternal antibiotic exposure and the increased risk of allergic inflammation in offspring. Supplementation of SCFAs in early life offers an attractive therapeutic strategy for the prevention of asthma in adults.

## Methods
### Mice
Germ-free (GF) C57BL/6J mice were supplied by Gempharmatech (Jiangsu, China). *Ifnar1*$^{-/-}$ mice were kindly provided by Dr. Chunsheng Dong Soochow University, China. *Rora*$^{fl/fl}$ *Il7r*$^{Cre}$ mice were kindly provided by Dr. Andrew N J McKenzie (Medical Research Council Laboratory of Molecular Biology, Cambridge, UK)[42]. *Gpr41*$^{-/-}$ mice were originally obtained from Cyagen Biosciences (Beijing, China), maintained on the C57BL/6J background, and C57BL/6J mice were purchased from Cyagen Biosciences. Mice were maintained in specific

pathogen-free conditions in the animal facility at Tianjin Medical University. Both genders were used. All mouse experiments were approved by the Institutional Animal Care and Use Committee of Tianjin Medical University.

### Preparation of cell suspensions from tissues
Single-cell suspensions were collected from lungs of adult mice[43]. Protocols for neonatal mice were similar but modified slightly. In brief, lungs from pups at different days after birth were flushed by cold PBS three times via the right ventricle of the heart before removal. Lungs were then removed, cut into small pieces, and homogenized in cold PBS at 200 mg lung tissue/mL. After centrifugation at 1000 g for 20 min, the supernatant was collected and stored at −80 °C for subsequent measurement. The pellet was resuspended in 3 mL collagenase type I (0.5 mg/mL in RPMI-1640 medium; Invitrogen) and incubated for 35 min at 37 °C in a shaker at 200 rpm. To isolate cells from the lungs, the digested tissue was homogenized and filtered through a 70 μm cell strainer. Red blood cells were lysed in ammonium chloride–potassium (ACK) buffer.

### Flow cytometric analysis and cell sorting
Single-cell suspensions were prepared from tissues as described above. To exclude dead cells, a LIVE/DEAD Fixable Aqua Dead Cell Staining Kit (ThermoFisher Scientific) was used. For ILC2 staining, cells were primarily labeled with a biotin-conjugated lineage antibody cocktail (including antibodies against CD3e, CD4, CD5, CD8a, Gr1, B220, NK1.1, CD11b, CD11c, Ter119, and TCR-β), and then stained with streptavidin-fluorochrome antibody and other surface antibodies including antibodies against CD45, CD90.2, and CD25. For staining of GATA3 and Ki67, cells were fixed and permeabilized (Foxp3/Transcription Factor Staining Buffer Set, Invitrogen), after staining with the antibodies against surface markers. For measuring intracellular IL-5 and IL-13 expression, cells were stimulated with 50 ng/mL PMA (Sigma-Aldrich), 1 μg/mL ionomycin (Sigma-Aldrich), and 1 μg/mL brefeldin A in complete RPMI-1640 medium for 4 h. Cells were labeled with antibodies to surface markers, fixed and permeabilized using an Intracellular Fixation and Permeability Kit (Invitrogen), and then stained with antibodies against IL-5 and IL-13. For analysis of p-STAT1 and p-STAT2, cells were fixed and permeabilized followed by staining with antibodies against p-STAT1 and p-STAT2. A CytoFLEX S flow cytometer (Beckman Coulter) and a MoFlo Astrios EQ (Beckman Coulter) cell sorter were used for sample acquisition and ILC2 sorting, respectively; the purity of ILC2 after sorting exceeded 90%. Data were analyzed with Flowjo V10.4. Antibodies used are listed in Supplementary Data-1.

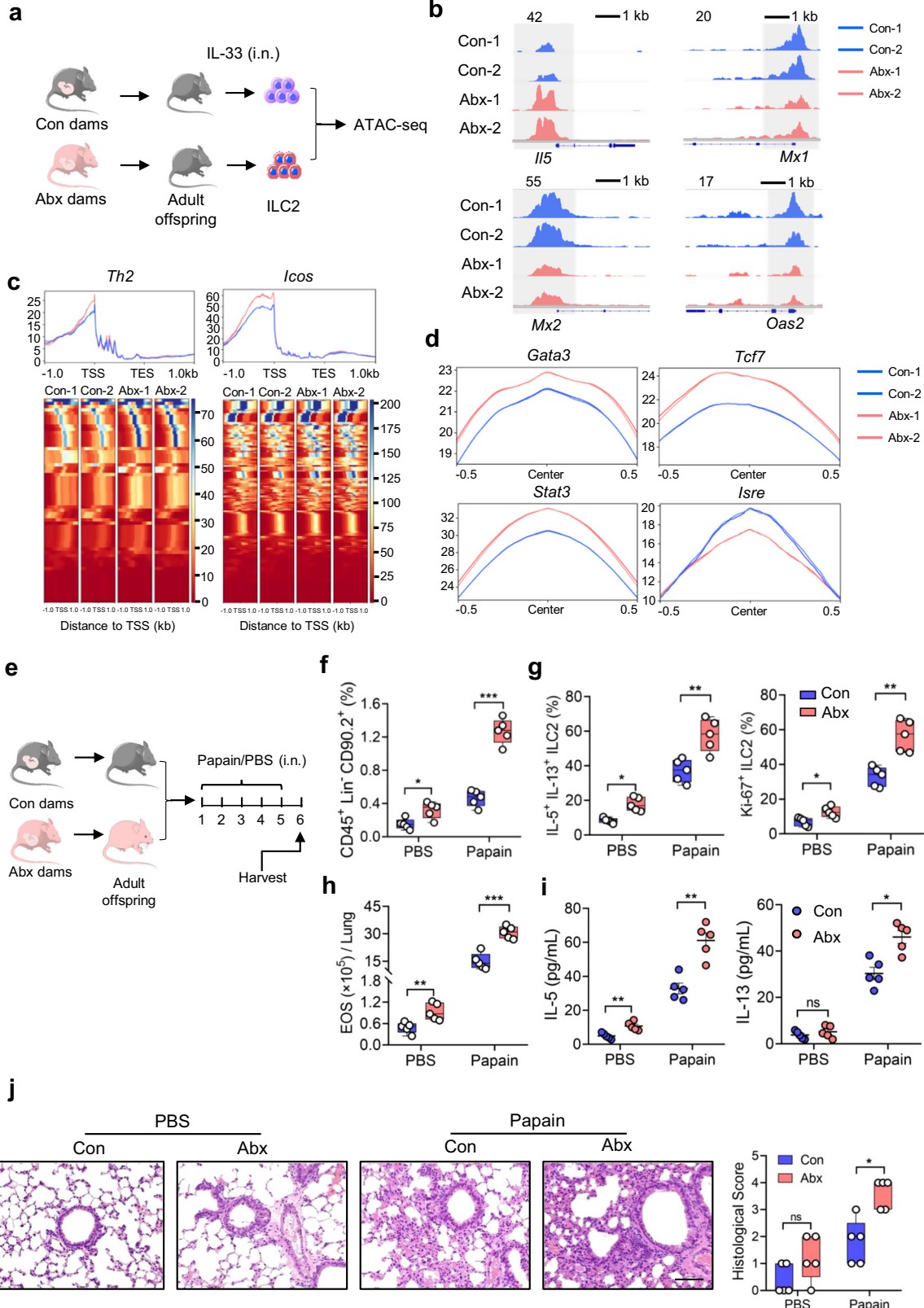

## Maternal antibiotic exposure

C57BL/6J pregnant mice were randomly supplemented with a broad-spectrum antibiotics cocktail, termed Abx, in their drinking water from gestation day 10 to day 14 to eliminate gut commensal microflora as described[13,14]. Abx consisted of ampicillin (1 g/L, Sigma), metronidazole (1 g/L, Solarbio), neomycin (1 g/L, Sigma), and vancomycin (0.5 g/L,

Sigma). Fecal samples were collected and confirm the efficiency of microbiota depletion by qRT-PCR.

## Fecal transplantation

Fresh intestinal contents were pooled from healthy control dams, weighed and suspended in sterile PBS (200 mg in 1 mL of PBS).

**Fig. 6 | Maternal antibiotic exposure induces epigenetic changes in ILC2s and exerts long-term effects on allergic inflammation in adult offspring. a–d** ILC2s from Abx adult offspring display distinct chromatin accessibility ($n = 2$). **a** Experimental design: adult offspring born to antibiotic exposed dams and the controls were challenged with IL-33 intranasally for 3 consecutive days (500 ng/mouse/day). Lung ILC2s were sorted for ATAC-seq analysis. **b** Integrated genome viewer snapshots of representative genes involved in Con and Abx ILC2s. Genomic regions differentially open in Con and Abx ILC2s are highlighted in gray boxes. Red dots indicate peaks that are more accessible in Abx ILC2s, and blue dots indicate those that are uniquely accessible in control ILC2 (fold change >2, FDR < 0.05). **c** Integrated genome viewer snapshots of representative genes involved in Th2 signaling and ICOS. **d** HOMER de novo motif analysis of multiple transcription factors important for ILC2 function. **e–j** Adult offspring born to Abx dams or controls were administered with papain or PBS intranasally for 5 consecutive days (n = 5). **e** Illustration of experimental model. Adult offspring born to Abx dams or controls were administered with papain or PBS intranasally for 5 consecutive days. Mice were analyzed on day 6. ILC2 phenotype and allergic inflammation in lungs were evaluated (**f–j**). The frequencies of ILC2s in lungs (**f**), IL-5$^+$IL-13$^+$ ILC2s, and Ki-67$^+$ ILC2s (**g**) and absolute numbers of eosinophils in lung (**h**) were analyzed by flow cytometry. (**i**): Amounts of IL-5 and IL-13 in BALF were measured by ELISA. **j** H&E staining of lung tissues (bar, 100 μm) and histological score. In Fig. 6i, the data are presented as the mean ± SEM values, by unpaired two-tailed Student's *t* test. For box plots, the data are shown as "Min to Max, show all points". For box plots, the midline represents the median; box represents the interquartile range (IQR) between the first and third quartiles, and whiskers represent the lowest or highest values within 1.5 times IQR from the first or third quartiles (**f, g, h, j**). *$P < 0.05$; **$P < 0.01$; ***$P < 0.001$; ****$P < 0.0001$ by unpaired two-tailed Student's *t* test or one-way ANOVA followed by Tukey-Kramer multiple-comparisons test. Data are presentative of 2-3 independent experiments (**f–j**). Statistical source data are provided in Source Data.

The solution was vortexed for 10 s, followed by centrifuge at 800 × g for 3 min to remove the large particles. Then the supernatants were collected and transplanted into Abx-treated pregnant dams by gavage as described previously[44].

## Allergic airway inflammation model

Adult mice were challenged with papain (20 μg in 40 μL PBS/mouse/day, Sigma) or PBS for 5 days intranasally. After the last challenge, the mice were sacrificed and BALF and lung tissues were analyzed. For butyrate treatment in the prenatal period, dams' drinking water was supplemented with butyrate (200 mM)[39] from embryonic day 14 (E14) to postnatal day 7. Post-weaning pups of Abx-exposed dams were given drinking water containing 200 mM butyrate from postnatal day 21 days for 2 weeks.

## Bacterial 16S rDNA amplicon and high-throughput sequencing

The CTAB/SDS method was used to extract total genomic DNA in feces from the distal colon of mice after administration of antibiotics (Abx) or PBS. 16S rDNA genes of the V4 region were amplified, using specific primers (F: 5′-GTG CCA GCM GCC GCG GTA A-3′; R: 5′-GGA CTA CHV GGG TWT CTA AT-3′) with a barcode. The NEBNext Ultra IIDNA Library Prep Kit was used for library construction and NovaSeq6000 was used for on-board sequencing. After read splicing and filtering, QIIME2 software was used to calculate the alpha diversity index and for beta diversity analysis, which revealed differences in species composition and community structure among samples. To determine the significance of differences in community structure between groups, the adonis and anosim functions in the QIIME2 software were used. To identify significantly different species at each taxonomic level (phylum, class, order, family, genus, species), R software (Version 3.5.3) was used for MetaStat and *T*-test analysis[45].

## Non-targeted metabolomic analysis

Non-targeted metabolomic analysis was based on liquid chromatography–mass spectrometry (LC-MS). Serum samples (100 μL, $n = 4$/group) were vortex-mixed with *X* μL prechilled 80% methanol/0.1% formic acid. The samples were then incubated on ice for 5 min and centrifuged at 15,000 g, 4 °C for 20 min. The supernatant was diluted with LC-MS grade water to its final concentration containing 53% methanol. The samples were centrifuged at 15,000 g, 4 °C for 20 min. Finally, the supernatant was injected into the LC-MS system[46,47]. Partial least squares discriminant analysis (PLS-DA) was performed using metaX[48]. The two-sided Wilcoxon rank-sum test was employed to identify differences, and the Benjamini-Hochberg method was applied for multiple testing correction. Metabolites with VIP >1 and $P < 0.05$ and fold change ≥2 or fold change ≤ 0.5 were considered to be differential metabolites. Volcano plots were used to filter metabolites of interest based on log$_2$ (fold change) and -log$_{10}$ (*P*-value) of metabolites by ggplot2 in R language. Raw data was shown in Supplementary Data-2.

## Short-chain fatty acid (SCFA) analysis

Pure standards of seven SCFAs (acetate, propionate, butyrate, valerate, isobutyrate, isovalerate and caproate) were prepared with diethyl ether to give 10 mixed standard concentration gradients. Serum samples (100 μL) were added to 50 μL15% phosphoric acid, then mixed with 10 μL internal standard (75 μg/mL isocaproate) and 140 μL ether. The mixture was centrifuged at 12,000 rpm at 4 °C for 10 min to obtain a supernatant which was separated on an Agilent HP-INNOWAX capillary column (30 m × 0.25 mm ID × 0.25 μm) gas chromatography system. A Thermo TRACE 1310-ISQ LT gas-mass spectrometer was used for mass spectrometry analysis. MSD ChemStation software was used to calculate the content of SCFAs in the samples[49].

## SMART-seq

A total of 1000 lung ILC2s were sorted and lysed in 5 μL lysis buffer ($n = 3$). The Smart-seq v4 Ultra Low Input RNA Kit (Clontech, Japan) was used to prepare a low-input library according to the kit's instructions. Cutadapt software (https://cutadapt.readthedocs.io/en/stable/, version: cutadapt-1.9) was used to remove reads that contained adapter contamination. After removing low-quality bases and undetermined bases, HISAT2 software (https://daehwankimlab.github.io/hisat2/, version: hisat2-2.0.4) was used to map reads to the genome (Homo sapiens Ensembl v96). The mapped reads were assembled using StringTie (http://ccb.jhu.edu/software/stringtie/,version: stringtie-1.3.4d). Expression levels of all transcripts were calculated using FPKM (total exon fragments/mapped reads (millions) × exon length (kb)). The DESeq2 package (version: DESeq2-1.40.0) was utilized to detect differentially expressed genes using the negative binomial distribution model. To control for multiple testing, the Benjamini-Hochberg method was employed. Differentially expressed mRNAs were selected with fold change > 2 or fold change <0.5 and adjusted $P < 0.05$. The GO terms (http://www.geneontology.org/) and KEGG pathways (http://www.genome.jp/kegg/) of these differentially expressed genes were then annotated. GSEA was analyzed using R packages (clusterProfiler version 3.10.1 and enrichplot version 1.2.0) as previously described[50]. Raw data was shown in Supplementary Data-3.

## ATAC-seq library preparation

Standard ATAC-seq was performed according to a published protocol[51]. Briefly, adult mice born to antibiotic exposed dams and those from healthy control adults were challenged with IL-33 intranasally for 3 consecutive days (500 ng/mouse/day). 50,000 ILC2s were sorted from lungs and were collected in 1.5 mL RNase-free tubes. Cells were centrifuged at 500 × g for 5 min at 4 °C, and the supernatant was removed without disturbing the cell pellet. The

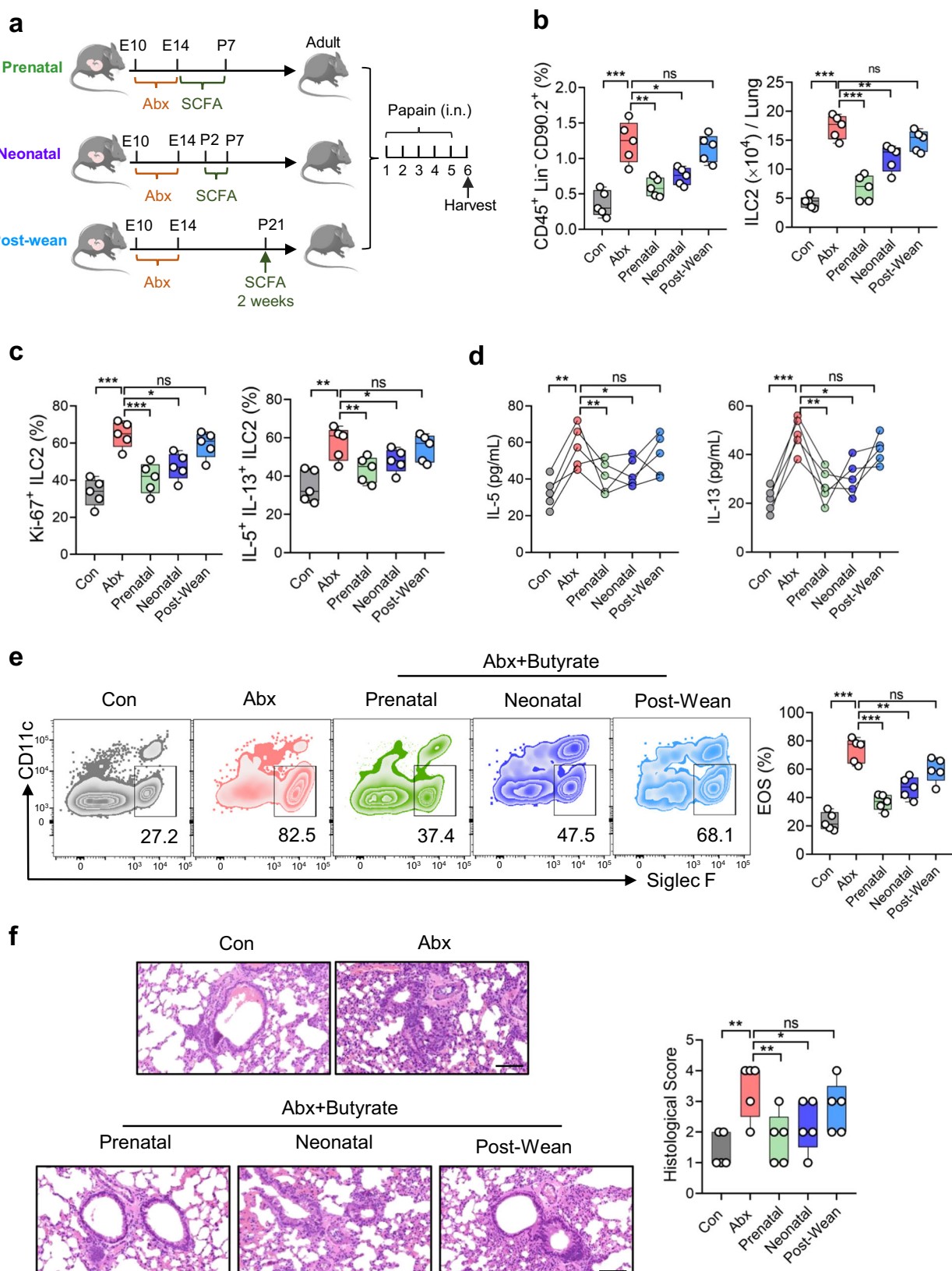

cells were resuspended in 50 μL chilled lysis buffer (10 mM Tris-HCl, pH 7.4, 10 mM NaCl, 3 mM MgCl₂, 0.5% NP-40, 0.1% Tween-20, and 0.01% digitonin)[52] and kept on ice for 10 min for further lysis. To collect nuclei, the cells were centrifuged at $500 \times g$ for 5 min at 4 °C, and the supernatant was removed. We then added mixture, which contains transposase (Vazyme TD501) and equimolar amounts of

two adapters, Adapter 1 and Adapter 2, and incubated the sample at 37 °C for 30 min to complete the transposition and connect the adapters. The final library was obtained by primer amplification, fragment length selection, and purification. Specific operation procedures and PCR conditions were provided in the instructions attached to the kit (TruePrep DNA Library Prep Kit V2 for Illumina).

**Fig. 7 | Therapeutic window of allergic airway inflammation in early life. a** Schematic diagram of butyrate supplementation at different periods. For Prenatal group, antibiotic-exposed dams were supplemented with butyrate (200 mM) in drinking water from embryonic day 14 (E14) to postnatal day 7 (P7) ($n = 5$); for Neonatal group, pups of Abx dams were intraperitoneally injected with butyrate daily from P2 to P7 ($n = 5$); for Post-Wean group, pups of Abx dams were supplemented with butyrate (200 mM) in drinking water from P21 for 2 weeks ($n = 5$). After butyrate supplementation, adult mice were administered with papain or PBS intranasally for 5 consecutive days. ILC2 phenotype and allergic inflammation in lungs were evaluated (**b**–**f**). **b**–**c** Frequencies and absolute numbers of ILC2s in lungs (**b**). Ki-67$^+$ ILC2s and IL-5$^+$ IL-13$^+$ ILC2s (**c**) were evaluated by flow cytometry. **d** Amounts of IL-5 and IL-13 in BALF were examined by ELISA. **e** Frequencies of eosinophils in BALF were analyzed by flow cytometry. **f** H&E staining of lung tissues (bars, 100 μm). In Fig. 7d, the data are presented as the mean ± SEM values, by unpaired two-tailed Student's *t* test. For box plots, the data are shown as "Min to Max, show all points". For box plots, the midline represents the median; box represents the interquartile range (IQR) between the first and third quartiles, and whiskers represent the lowest or highest values within 1.5 times IQR from the first or third quartiles (**b**, **c**, **e**, **f**). *$P < 0.05$; **$P < 0.01$; ***$P < 0.001$; ****$P < 0.0001$ by unpaired two-tailed Student's *t* test. Data are presentative of 2-3 independent experiments (**b**–**f**). Statistical source data are provided in Source Data.

ATAC-seq data were collected using Illumina NovaSeq with 2 × 150 bp paired-end run.

## ATAC-seq data processing and analysis
To call ATAC-seq peak data, the nf-core atacseq pipline (version 1.2.2)[53] was applied with parameters (--genome mm10 --narrow_peak). The procedures could be simplified to the followings. Adapter trimming was accomplished by Trim Galore (version 0.6.5) (https://github.com/FelixKrueger/TrimGalore). Next, clean reads sequence alignment was performed using the mouse reference genome (mm10) with BWA (version 0.7.17)[54] and then filtered with default parameters using SAMtools (version 1.10) as well as Picard (version 2.23.1) (https://broadinstitute.github.io/picard/)[55]. MAnorm2[56] was applied to normalize data from different samples as well as identify the specific peaks. To control for multiple testing, the Benjamini-Hochberg method was employed. As a consequence, 48788 peaks (41727 from Abx Group and 45352 from Con Group) were filtered for the downstream analysis. We annotated all peaks by using HOMER (version 4.11)[57]. As to pathway analysis, signaling pathway-related genes were chosen from the Molecular Signatures Database (MSigDB, version 7.5.1)[58] meanwhile their transcripts were collected from UCSC Table Browser (mm10, NCBI RefSeq). Based on the commands bamCoverage (--binSize 10 --normalizeUding RPGC), computeMatrix (--upstream 1000 --downstream 1000), plotHeatmap, and plotProfile, deepTools (version 3.4.3)[59] was applied to compare the chromatin open state around transcription start site (TSS) between Abx group and the control. To compare the chromatin open state at the sites of motif occurrences, deepTools (version 3.4.3) was also used. When it came to visualize the tracks, the Integrative Genomics Viewer (IGV, version 2.12.3)[60] based on the 1x depth (reads per genome coverage, RPGC) normalization was used to make the signals from target genes between Abx group and the control comparable. Primary data was listed in Supplementary Data-4.

## ILC2 culture in vitro
ILC2s (1 × 10$^4$) from lungs were sorted and cultured in the presence of 100 ng/mL IL-33, 20 ng/mL IL-2, and 20 ng/mL IL-7 for 3 days[27]. The supernatants were collected for IL-5, and IL-13 measurement by ELISA. The concentrations of indicated treatment agents were 2 mM butyrate, 1 mM AR420646[24], 0.5 mM 4-CMBT[25], 5 μM Fludarabine[21], and 0.4 mg/mL hydrocortisone[22].

## Quantitative RT-PCR (qRT-PCR)
Total mRNA from sorted ILC2s was extracted by TRIzol reagent (Solarbio, Beijing) and reverse transcribed using a cDNA synthesis kit (Takara, Japan). The mRNA expression of target genes was analyzed by RT-qPCR. Primer sequences are listed in Supplementary Data-5.

## Lung histology
Lung tissues were fixed in 4% paraformaldehyde for 24 h. Tissue sections were prepared and stained with hematoxylin-eosin (H&E) to evaluate inflammation. Regarding the scoring of lung inflammation, tissue sections were scored by a histopathologist in a blinded manner based on the established scoring standard: 0, normal; 1, very mild; 2, mild; 3, moderate; 4, severe[61]. An increment of 0.5 was used when the inflammation fell between two levels[62]. Three fields were selected randomly for scoring using a Leica microscope by two treatment-blind pathologists independently[63].

## ELISA
BALF or lung homogenates from mice, or culture supernatants from ILC2s, were collected. ELISA was applied to measure the amounts of IFNβ (Cloud Clone, Wuhan), IL-5 (Invitrogen), and IL-13 (Invitrogen).

## Adoptive transfer of ILC2s
Adoptive transfer of ILC2s was performed[64]. In brief, lung ILC2s were sorted from neonatal mice (postnatal day 7) born to antibiotic-exposed dams and littermate controls, and were then intravenously transferred into NCG mice (1.5 × 10$^4$ cells/mouse), which were then challenged intranasally with IL-33 (500 ng/mouse, daily) or PBS for 3 consecutive days. Mice were sacrificed and analyzed 24 h after the last challenge.

## Quantification and statistical analysis
Statistical details and methods for each experiment can be found in figure legends. We presented data as mean ± SEM. unless otherwise indicated. Statistical analysis was performed using GraphPad Prism 8.0. All *P*-values are indicated in the figures and figure legends.

## Reporting summary
Further information on research design is available in the Nature Portfolio Reporting Summary linked to this article.

## Data availability
The SMART-seq data generated in this study have been deposited in the Gene Expression Omnibus (GEO) under accession code GSE231887. ATAC-seq data have been deposited into GEO under the series entry number of GSE242246 and can be downloaded from the link below: https://www.ncbi.nlm.nih.gov/geo/query/acc.cgi?acc=GSE242246. All other data are available in the article and its Supplementary files or from the corresponding author upon request. Source data are provided with this paper.

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

## Acknowledgements

This work was supported by the following grants to J.Z.: National Natural Science Foundation of China (No.81925018, 82130049). the Start-up Funding for High-level Talents of Tianjin Medical University, Key Project of Tianjin Natural Science Foundation (20JCZDJC00670). It was also supported by the following grant to H.X.: The Science & Technology Development Fund of Tianjin Education Commission for Higher Education (2020KJ194).

## Author contributions

J.Z. conceived and supervised this study. Z.Y. and D.H. co-supervised this study. H.X. and Z.C. performed the experiments. X.Y. and X.F. conducted computational analysis with supervision from D.H. and M.J.L., H.L., L.Zhu., J.C., L.Zhang. and P.Z. participated in animal experiments. Y.Y. and Q.L. provided suggestions and advice in project design and manuscript revision. J.Z. wrote the manuscript with inputs from all authors.

## Competing interests

The authors declare no competing interests.
