## [Peer Review File · Nature Communications]

Maternal antibiotic exposure enhances ILC2 activation in neonates via downregulation of IFN1 signalingREVIEWER COMMENTS

Reviewer #1 (Remarks to the Author):

Xu, H., et al present this manuscript observing the effect of maternal microbiota on neonatal ILC2 activity and its role in allergic inflammation. They utilized several techniques including FMT, mice knockouts, and SPF and GF mice to demonstrate that antibiotic administration during pregnancy contributed to increased neonatal ILC2 and increase in allergic inflammation as demonstrated by increased eosinophils, pathological lung remodeling, and increased IL-5 and IL-13. Additionally using bioinformatic techniques such as RNAseq and ATAC-seq they were able to elucidate a mechanism involving decreased butyrate production in supporting increased ILC2 activity. Overall, this study is interesting, however, some additional experiments and modifications are required to improve the manuscript and to support the claim.

1. What is the meaning of ILC2 number /mL that appeared in Fig S1C and Fig2F? The more appropriate way is to report the number of ILC2/lung. Likewise, in Fig S1F & Fig1D & Fig1I & Fig2B): What is the unit of EOS number?

2. Fig3B-E: There might be a significant difference between C>C and C>A in terms of ILC2 number and percentage, Ki67, IL13, and EOS count. If the results appeared to be significant, the results suggest that the effect of the maternal microbiota on neonatal ILC2s is not merely determined postnatally and both prenatal and postnatal items could affect the airway inflammation.

3. Figure 4B: cell cycle pathways are highly significant but not highlighted, authors should elaborate on why are T-cell receptor signaling showing up.

4. Page 12: "Gene profiling showed that the transcripts of Gpr41 were most abundant in neonatal ILC2s compared with Gpr41 and Gpr109a, Hdac9 (Figure S4E)." The authors may intend to write Gpr43. Many unnecessary bold and underline words and sentences throughout the manuscript.

5. Page 12: “AR420626, a selective agonist of GPR41 (Sepahi et al. 2021), efficiently diminished neonatal ILC2 responses, whereas GPR43 agonist 4-CMTB (Thio et al. 2018) displayed no effect (Figure S4F).” FigS4F is about the evaluation of Gpr41 mRNA expression following the addition of its agonist, AR420626. It is not about the evaluation of ILC2 responses following AR420626 addition. Also, the ILC2 response was not compared following the addition of AR420626 or 4-CMTB.

6. Why the addition of Gpr41 agonist leads to Gpr41 mRNA upregulation? Please elaborate.

7. Page 12, misinterpretation of data: “In addition, the GPR41 antagonist polyhydroxy butyrate (PHB) (Won et al. 2013) almost completely abolished the effect of butyrate on ILC2s (Figure S4G).” According to the results of FigS4G, adding GPR41 antagonist (PHB) did not “completely abolish” the effect of butyrate on ILC2s. The decrease is statistically significant, but still has a long way to reach the state in which no butyrate was added.

8. Page 13, misinterpretation of data: “These observations indicate that maternal antibiotic exposure induces persistent epigenetic changes in ILC2s in offspring.” How did you claim that these epigenetic changes are persistent? Please elaborate.

9. In Figure 3, the functional activation of ILC2 is alleviated by cross-fostering, but there is an insufficient explanation regarding the mechanism of action. The authors need to show the relationship between butyrate or SCFA in breast milk to show consistency with subsequent studies. Also, most importantly, the authors need to address or consider the possibility that microbiota in the feces or skin of control dams could have been orally transferred to and affected the neonates.

10. Figure 7, it is necessary to separate SCFA administration between fetal and neonatal periods or to show that there is no difference between placental and breast milk transfer, considering the clinical situation. The SCFA dosage may need to be adjusted in prenatal and post-wean treatment.

11. There is no description of specific evaluation methods for histological scoring.

12. The discussion is weak and should be enriched by discussing and comparing related studies.

Reviewer #2 (Remarks to the Author):

In their manuscript XU et al. described the impact of maternal microbiota on mouse lung ILC2 functions through a mechanism that mobilized type I IFN pathway and bacteria derived butyrate. The study is based on antibiotic mediated microbiota depletion and the use of full or conditional gene targeted KO mice. The deprivation of microbiota during pregnancy results in development of lung type 2 inflammation associated with ILC2. This process is associated with reduced IFN γ signalling, the modulation of IFN I being mimicked by butyrate. Long term effect being associated with reciprocal epigenetic regulation at the type 2 cytokine and type 1 IFN loci in ILC2. FACS gating strategy for ILC analysis and isolation is accurately documented. In conclusion, metabolite derived from maternal microbiota impact ILC2 through IFN I pathway. The topic is of interest as well as the conclusions.

The study emphasize the role of maternal microbiota-derived butyrate, and Fig S1B and Fig S3 analyzed maternal microbiota diversity, unclear for alpha diversity, and more for beta diversity. The main issue about the conclusion is whether the treatment during pregnancy affects the microbial colonization of the pups in terms of alpha and beta diversity (intestine and lungs). Page 10 Line 14, it is written “we next evaluated the changes of bacterial composition in pups...”. However, Fig S3 legend mentions “composition of maternal microbiota”. The assumption that the composition will be the same in dams and pups need to be demonstrated, as microbial colonization in pups will directly affect ILC2 development. The cross fostering experiments do not help to conclude. This can obviously have a major impact on ILC2 development, and may explain the epigenetic regulation at distance once adults (Figure 6). The nature of initial imprinting would be completely different. The microbial analysis of the offsprings needs to be clearly assessed.

The assumption that SCFA levels are directly related to the exposure of a lower abundance of SCFA-producing species, namely *Lactobacillus*, in pages 10/11 remains to be

demonstrated. Mono-colonization of GF mice with this strain could provide such demonstration, and direct impact on ILC2.

There is a lot of a confusion about IFN I in the present study, and its clear role needs to be experimentally investigated. According to the data, IFN I is expressed by ILC2, influenced the development of ILC2 in numbers, increased the signalling of IFN I pathway and decreased the type 2 cytokine response in ILC2. Fig 5C shows that butyrate induced IFN I by ILC2 in vitro in the presence of IL-33. Fig5Q shows in vivo induction of IFN I in lung homogenates, without demonstration that this involves ILC2. Whether this corresponds to a self-regulation? bystander one, or both is important. Are ILC2 natural IFN I producers or in which conditions? Full IFNARko is not helpful, and ifnar1-floxed mice have been generated a while ago and would help to clarify the role of IFN I role. Direct induction of IFN-beta by bacteria has been extensively documented. For instance, alveolar macrophages can be a major source of this cytokine in the context of bacterial exposure in the lungs.

The paper by Schneider et al (2019) could be better discussed in light of the results provided.

The current data could be discussed from the perspective of type 1, type 2, type 3 and regulatory balances. For instance, IFN-beta is alternative to IL-12 to favor type 1 immune responses. Fig S1 shows at the intestinal level an increase of ILC2 in Abx mice and a decrease of ILC3. Whether different type of ILC were found in the lungs is not mentioned.

The "ILC2 KO" mouse is not well referenced in the manuscript, sometimes as IL7rCreRorafl/fl in the text and as IL17CreRorafl/fl in figure legends and methods. This need to be corrected, and the original publication of these mice to be added for the sake of clarity. In general, the different mouse models used in the study need to be detailed.

The reference list do not follow the format of the journal. The referencing format mixes "authors+year" in the text and numbers in brackets in the list, which is not convenient and not correct. It mixes full first name and abbreviations, dates of publication etc. This can lead to errors, such as "Viver et al" in the text which are 'Vivier et al' in the reference list. A

Careful revision of the references are needed.

Reviewer #3 (Remarks to the Author):

This study sheds new light on the complex relationship between maternal microbiota and type 2 innate immunity in infancy, revealing important cellular and molecular mechanisms for the causal relationship between maternal antibiotic exposure and increased risk of allergic inflammation in offspring. The design of this study is relatively novel and of profound clinical significance, but the reviewers still have doubts about the results of the study, which requires the author to revise and explain.

Major

1. The ILC2 cells selected in this study were the main experimental objects, and the authors did not provide the most direct experimental evidence to prove the important role of ILC2 in this study. Moreover, the proportion of ILC2 in mouse lung tissue makes it difficult to extract primary cells, and it is necessary to prove the purity of the obtained cells and the possibility of differentiation.
2. It needs to be confirmed whether the NCG mice selected by the authors will cause errors in experimental results due to metabolic disorders.
3. The authors describe in detail the changes in fetal rats associated with symptoms as they grow older, but do not provide any explanation of the mechanism. In the whole study, logical sorting and direct evidence are a little weak.
4. Fig 7 : Whether the authors considered the effect of butyrate treatment on pregnant mice without antibiotic treatment. How to confirm the concentration, use time and dosage of butyrate. What is the key effect of butyrate on pregnant mice? What is the metabolism and distribution of butyrate in pregnant mice?

Minor

1. The pulmonary edema and the changes of respiratory function in mice can show the injury of lung tissue more comprehensively.
2. The authors can select markers of ILC-2 cells for immunofluorescence or immunohistochemical staining to show its distribution in lung tissue.

REVIEWER COMMENTS

Reviewer #1 (Remarks to the Author):

Xu, H., et al present this manuscript observing the effect of maternal microbiota on neonatal ILC2 activity and its role in allergic inflammation. They utilized several techniques including FMT, mice knockouts, and SPF and GF mice to demonstrate that antibiotic administration during pregnancy contributed to increased neonatal ILC2 and increase in allergic inflammation as demonstrated by increased eosinophils, pathological lung remodeling, and increased IL-5 and IL-13. Additionally using bioinformatic techniques such as RNAseq and ATAC-seq they were able to elucidate a mechanism involving decreased butyrate production in supporting increased ILC2 activity. Overall, this study is interesting, however, some additional experiments and modifications are required to improve the manuscript and to support the claim.

1. What is the meaning of ILC2 number /mL that appeared in Fig S1C and Fig2F? The more appropriate way is to report the number of ILC2/lung. Likewise, in Fig S1F & Fig1D & Fig1I & Fig2B): What is the unit of EOS number?

We corrected the units of ILC2 numbers to the absolute cell counts per lung in Figure S1C and Figure 2F, represented as ILC2 number ($\times 10^4$) / Lung. Similarly, the unit of eosinophils (EOS) in Figure S1F, 1D, 1I, 2B, is the absolute cell counts of EOS per lung. We clarified the information in Figure Legends.

2. Fig 3B-E: There might be a significant difference between C>C and C>A in terms of ILC2 number and percentage, Ki67, IL13, and EOS count. If the results appeared to be significant, the results suggest that the effect of the maternal microbiota on neonatal ILC2s is not merely determined postnatally and both prenatal and postnatal items could affect the airway inflammation.

We agree with the reviewer. We reanalyzed the data and there were significant differences between C→C and C→A groups in terms of the frequencies of ILC2, Ki67⁺ILC2, EOS, as well as the amounts of IL-13 in lungs (**Figure 3B-3E**). These

observations indicate that the contribution of postnatal factors to the observed higher ILC2 responses born to Abx dams could not be excluded, such as the changes in microbiota of pups. Another possibility is that microbiota in the feces or skin of control dams could have been orally transferred to and affected the neonates. We provided this in the Discussion (the first paragraph of page 17).

3. Figure 4B: cell cycle pathways are highly significant but not highlighted, authors should elaborate on why are T-cell receptor signaling showing up.

Cell cycle pathways were upregulated in ILC2s from Abx group, which supports their higher proliferative status. It has been reported that there is direct interaction between ILC2 and CD4 T cell responses. ILC2s cells may directly interact with CD4 T through MHCII-peptide-TCR. ILC2s could provide costimulatory signals to T cells. Activated ILC2 could promote Th2 responses during allergic inflammation (Reviewed by Gurram RK, et al. Cell Mol Immunol. 2019 PMID: 30792500). These may explain the higher T cell receptor signaling in ILC2s from Abx dams. We provided this information in Results (line 4-6 on page 9).

4. Page 12: “Gene profiling showed that the transcripts of Gpr41 were most abundant in neonatal ILC2s compared with Gpr41 and Gpr109a, Hdac9 (Figure S4E).” The authors may intend to write Gpr43. Many unnecessary bold and underline words and sentences throughout the manuscript.

We are sorry for this mistake. We corrected it. We also double checked the revised manuscript.

5. Page 12: “AR420626, a selective agonist of GPR41 (Sepahi et al. 2021), efficiently diminished neonatal ILC2 responses, whereas GPR43 agonist 4-CMTB (Thio et al. 2018) displayed no effect (Figure S4F).” FigS4F is about the evaluation of Gpr41 mRNA expression following the addition of its agonist, AR420626. It is not about the evaluation of ILC2 responses following AR420626 addition. Also, the ILC2 response was not compared following the addition of AR420626 or 4-CMTB.

Sorry for this mistake. We reorganized the data and corrected the statement. The amounts of type 2 cytokines measured by ELISA was updated in **Figure S4D**. The results indicate that GPR41 agonist AR420626 reduced ILC2 responses, GPR43 agonist 4-CMTB displayed no noticeable effect. We also compared ILC2 response following the addition of AR420626 or 4-CMTB. ILC2 responses were lower in AR420626 group as compared with 4-CMTB. We deleted the original Figure S4F about Gpr41 mRNA expression.

6. Why the addition of Gpr41 agonist leads to Gpr41 mRNA upregulation? Please elaborate.

Based on literatures (Mikami, *et al.* Ther Adv Med Oncol. 2020, PMID: 33014144; Yin, *et al.* Int J Biol Macromol. 2020, PMID: 32229214), GPR41 agonist AR420626 could reduce HDAC activity and upregulate GPR41 mRNA expression. Due to the limitation of space, we deleted this data.

7. Page 12, misinterpretation of data: “In addition, the GPR41 antagonist polyhydroxy butyrate (PHB) (Won et al. 2013) almost completely abolished the effect of butyrate on ILC2s (Figure S4G).” According to the results of FigS4G, adding GPR41 antagonist (PHB) did not “completely abolish” the effect of butyrate on ILC2s. The decrease is statistically significant, but still has a long way to reach the state in which no butyrate was added.

We agree with the reviewer's concern. We tuned down the statement to “In addition, coadministration of GPR41 antagonist polyhydroxy butyrate (PHB) (Won et al. 2013) significantly reduced ILC2 responses as compared with butyrate alone (**Figure S4E**)”.
(page 12, line 15-17)

8. Page 13, misinterpretation of data: “These observations indicate that maternal antibiotic exposure induces persistent epigenetic changes in ILC2s in offspring.” How did you claim that these epigenetic changes are persistent? Please elaborate.

We collected lung ILC2s from adult offspring born to Abx dams and control dams,

and ATAC sequencing (ATAC-seq) was performed to evaluate the chromatin accessibility. ILC2s from Abx group showed enhanced chromatin accessibility of ILC2 hallmark genes such as *Ii5* and *Icos*, whereas ILC2s from control group exhibited higher chromatin accessibility of IFN-stimulated genes (ISGs) such as *Mx1*, *Mx2* and *Oas2* (**Figure S5C & 6B**). These observations were consistent with mRNA transcriptional analysis of neonatal ILC2s by SMART-seq (**Figure 4**). Based on these observations, the changes of ILC2s in offspring in response to maternal antibiotic exposure may be long-lasting, although direct evidence is needed to prove the epigenetic changes are persistent. We understand the reviewer's concern, and we tuned down the statement to "These observations indicate that maternal antibiotic exposure induced epigenetic changes in ILC2s in adult offspring." (the last 2 line of page 13)

9. In Figure 3, the functional activation of ILC2 is alleviated by cross-fostering, but there is an insufficient explanation regarding the mechanism of action. The authors need to show the relationship between butyrate or SCFA in breast milk to show consistency with subsequent studies. Also, most importantly, the authors need to address or consider the possibility that microbiota in the feces or skin of control dams could have been orally transferred to and affected the neonates.

We collected breast milk from the control and antibiotic-exposed mother mice at 7 days postpartum, followed by targeted metabolomics to evaluate the amounts of SCFAs (**A-B**). Consistently, the amounts of butyrate were significantly reduced in breast milk from antibiotic-exposed mother as compared to the control (B). The data was updated in **Figure 5C**. Regarding the possibility that microbiota in the feces or skin of control dams could have been orally transferred to and affected the neonates, we agree with the reviewer that this possibility could not be excluded. We tuned down the statement and revised the title of Figure 3 to "Breast milk derived factors contribute to the effects of maternal microbiota on neonatal ILC2s" (page 7). We also provided explanation in Discussion (page 17 the 1st paragraph).

(A) Principal component analysis (PCA) of the targeted metabolomics from breastmilk between Abx-treated dams and control dams (n=3). (B) The amounts of distinct SCFAs in breast milk determined by targeted metabolomics analysis (n=3).

10. Figure 7, it is necessary to separate SCFA administration between fetal and neonatal periods or to show that there is no difference between placental and breast milk transfer, considering the clinical situation. The SCFA dosage may need to be adjusted in prenatal and post-wean treatment.

As required by the reviewer, we butyrate administration was performed at neonatal period (postnatal day 2 to day 7), The protective effects of butyrate on allergic airway inflammation in adult offspring were similarly observed. The data were updated in Figure 7. The dosage of SCFAs we used was based on previous study (Yu, et al. *Pharmacol Res.* 2020. PMID: 32679183), no adverse effects were observed in mice.

11. There is no description of specific evaluation methods for histological scoring.

Regarding the scoring of lung inflammation, tissue sections were scored by a histopathologist in a blinded manner based on the established scoring standard: 0, normal; 1, very mild; 2, mild; 3, moderate; 4, severe. An increment of 0.5 was used when the inflammation fell between two levels. Three fields were selected randomly for scoring using a Leica microscope by two treatment-blind pathologists independently (Wallrapp, et al. *Nature.* 2017. PMID:28902842; McKay, et al. *Journal of immunology.* 2004. PMID: 14978092; He, et al. *Cell reports.* 2019. PMID: 31775040). We included the detailed information in Methods (page 34-35, Lung histology).

12. The discussion is weak and should be enriched by discussing and comparing related studies.

We have substantially improved the Discussion by comparing the studies related to our observations.

Reviewer #2 (Remarks to the Author):

In their manuscript XU et al. described the impact of maternal microbiota on mouse lung ILC2 functions through a mechanism that mobilized type I IFN pathway and bacteria derived butyrate. The study is based on antibiotic mediated microbiota depletion and the use of full or conditional gene targeted KO mice. The deprivation of microbiota during pregnancy results in development of lung type 2 inflammation associated with ILC2. This process is associated with reduced IFN I signalling, the modulation of IFN I being mimicked by butyrate. Long term effect being associated with reciprocal epigenetic regulation at the type 2 cytokine and type 1 IFN loci in ILC2. FACS gating strategy for ILC analysis and isolation is accurately documented. In conclusion, metabolite derived from maternal microbiota impact ILC2 through IFN I pathway. The topic is of interest as well as the conclusions.

The study emphasize the role of maternal microbiota-derived butyrate, and Fig S1B and Fig S3 analyzed maternal microbiota diversity, unclear for alpha diversity, and more for beta diversity. The main issue about the conclusion is whether the treatment during pregnancy affects the microbial colonization of the pups in terms of alpha and beta diversity (intestine and lungs). Page 10 Line 14, it is written “we next evaluated the changes of bacterial composition in pups...”. However, Fig S3 legend mentions “composition of maternal microbiota”. The assumption that the composition will be the same in dams and pups need to be demonstrated, as microbial colonization in pups will directly affect ILC2 development. The cross fostering experiments do not help to conclude. This can obviously have a major impact on ILC2 development, and may explain the epigenetic regulation at distance once adults (Figure 6). The nature of initial

imprinting would be completely different. The microbial analysis of the offsprings needs to be clearly assessed.

The assumption that SCFA levels are directly related to the exposure of a lower abundance of SCFA-producing species, namely *Lactobacillus*, in pages 10/11 remains to be demonstrated. Mono-colonization of GF mice with this strain could provide such demonstration, and direct impact on ILC2.

Responses to the above points:

Regarding the point “Fig S1B and Fig S3 analyzed maternal microbiota diversity, unclear for alpha diversity, and more for beta diversity”, we analyzed the fecal contents of control and antibiotic exposure groups. Antibiotic exposure reduces beta diversity, but as the reviewers mention the difference is small relative to alpha diversity (**Supplemental Figure 3A & 3B**). Alpha diversity is defined as species richness on the local or habitat scale, and beta diversity is defined as the difference in the types of species found in different areas of alpha diversity (Vavrek, et al. *Proc Natl Acad Sci U S A*. 2010. PMID: 20404176). Alpha diversity, as indicated by the abundance-based coverage estimator (ACE), Chao1, Shannon indices, did not display differences between the Abx and control groups. Differences were observed only for beta diversity.

Regarding the point “Page 10 Line 14, it is written “we next evaluated the changes of bacterial composition in pups...”. However, Fig S3 legend mentions “composition of maternal microbiota”. Sorry for the mistake, we corrected it: “we next evaluated the changes of bacterial composition in **dams** with or without antibiotic exposure”.

Regarding the point “the assumption that the composition will be the same in dams and pups need to be demonstrated, as microbial colonization in pups will directly affect ILC2 development...The microbial analysis of the offsprings needs to be clearly assessed”, the intestinal microbiota of the pups was evaluated by 16sRNA sequencing. Results showed that the composition of microbiota was different between dams and pups (**Figure S3C-S3F**), which was consistent with previous reports (Reyman, et al. *Nat Commun*. 2019. PMID: 31676793). In line with the observations from dams, pups

delivered from Abx-treated dams also displayed lower abundance of butyrate-producing Firmicutes phylum and Lactobacillaceae family (A-B). We provided the data from pups in **Figure S3E-S3F** (the 1st paragraph of page 11).

(A): Relative abundance of phylum level in pups (n=5). (B): Relative abundance of family level in pups (n=5).

Regarding the point “The assumption that SCFA levels are directly related to the exposure of a lower abundance of SCFA-producing species, namely Lactobacillus, in pages 10/11 remains to be demonstrated. Mono-colonization of GF mice with this strain could provide such demonstration, and direct impact on ILC2”, we agree with the reviewer that other SCFA-producing species may also contribute to SCFAs production in dams and the pups they delivered in this study, in addition to Lactobacillus. In the therapeutic strategy in Figure 7, we therefore used butyrate supplementation, rather than Lactobacillus.

There is a lot of a confusion about IFN I in the present study, and its clear role needs to be experimentally investigated. According to the data, IFN I is expressed by ILC2, influenced the development of ILC2 in numbers, increased the signalling of IFN I pathway and decreased the type 2 cytokine response in ILC2. Fig 5C shows that butyrate induced IFN I by ILC2 in vitro in the presence of IL-33. Fig5Q shows in vivo induction of IFN I in lung homogenates, without demonstration that this involves ILC2. Whether this corresponds to a self-regulation? bystander one, or both is important. Are ILC2 natural IFN I producers or in which conditions? Full IFNARko is not helpful, and ifnar1-floxed mice have been generated a while ago and would help to clarify the role

of IFN I role. Direct induction of IFN-beta by bacteria has been extensively documented. For instance, alveolar macrophages can be a major source of this cytokine in the context of bacterial exposure in the lungs.

We understand the reviewer's concerns about the role of IFN I signaling and ILC2 responses. ILC2s expressed IFN1 receptor *Ifnar1* (Duerr, et al. Nat Immunol. 2016. PMID: 26595887). Previous studies have demonstrated that IFN1 signaling negatively regulated ILC2 responses and alleviated the pathogenesis of allergic airway inflammation (Duerr, et al. Nat Immunol. 2016. PMID: 26595887; Block, et al. Nat Immunol. 2022. PMID: 36411381). Administration of IFN- α to neonatal mice prevented the development of allergic airway inflammation in IFNAR1 dependent manner (Wu, et al. Cell Mol Immunol. 2020. PMID: 31853001). However, whether IFN1 signaling regulates ILC2 in paracrine or autocrine manner, still remains to be investigated. In this study, we found that IFN1 signaling in ILC2s from pups was downregulated in response to maternal antibiotic exposure, administration of IFN- β suppressed the production of effector cytokines from ILC2 both *in vitro* and *in vivo*. In addition, upregulation of *Ifnb1* was observed in cultured ILC2s under HSV infection, the levels of IFN- β in the culture supernatants from ILC2s were comparable to those of macrophages in response to HSV infection (**Figure S2D-S2E**). These results indicate that IFN1 signaling may regulate ILC2 *via* autocrine manners. We also updated the explanation in Discussion (page 18, 1st paragraph).

(A): *qRT-PCR* analysis of *Ifnb1* mRNA in neonatal lung ILC2 treated with HSV (MOI, 10) for 12 hours ($n=3$). (B): Neonatal lung ILC2s were infected with HSV (MOI, 10) to detect secreted IFN- β in culture supernatants by ELISA ($n=3$). macrophages from neonatal spleen were used as positive control.

The paper by Schneider et al (2019) could be better discussed in light of the results provided.

We cited this paper and discussed it better.

The current data could be discussed from the perspective of type 1, type 2, type 3 and regulatory balances. For instance, IFN-beta is alternative to IL-12 to favor type 1 immune responses. Fig S1 shows at the intestinal level an increase of ILC2 in Abx mice and a decrease of ILC3. Whether different type of ILC were found in the lungs is not mentioned.

Although different subsets of ILCs were found in lungs, ILC2s play an essential role in the initiation of progression in the course of type 2 allergic airway inflammation (Spits. et al. Nat Rev Immunol. 2022, PMID: 35354980). That's why we focused on ILC2 in this study. The role of ILC3 in respiratory diseases is less studied, they play an important role in tissue homeostasis, infection and inflammation in gut via secretion of type 3 cytokine IL-17 and IL-22 Ardain, et al. Front Immunol. 2019. PMID: 30761149; Siracusa, et al. Nat Immunol. 2023. PMID: 37580603). In this study, we did not observe significant changes in the frequencies of ILC1 and ILC3 in intestines from neonates born of Abx and control dams. The potential effects of maternal microbiota on the function of other ILC subsets in offspring deserves further investigation, such as utilization of infectious model and colitis model. This will broaden our understanding about the comprehensive roles of maternal microbiota in the development of innate immune system in early life. We included in Discussion (page 17, 2nd paragraph).

The “ILC2 KO” mouse is not well referenced in the manuscript, sometimes as IL7rCreRorafl/fl in the text and as IL17CreRorafl/fl in figure legends and methods. This need to be corrected, and the original publication of these mice to be added for the sake of clarity. In general, the different mouse models used in the study need to be detailed.

We used *Rora^{fl/fl}Il7r^{Cre}* (ILC2 deficient mice from Dr. Andrew N J McKenzie), to evaluate the contribution of ILC2s to the observed phenotype. This strain was made by

Dr. Andrew N J McKenzie and was used by other groups (Oliphant, *et al.* Immunity. 2014. PMID: 25088770; Moral, *et al.* Nature. 2020. PMID: 32076273). We provided the detailed information and references in the Methods (page 27).

The reference list do not follow the format of the journal. The referencing format mixes “authors+year” in the text and numbers in brackets in the list, which is not convenient and not correct. It mixes full first name and abbreviations, dates of publication etc. This can lead to errors, such as “Viver et al” in the text which are ‘Vivier et al’ in the reference list. A careful revision of the references is needed.

We formatted the references based on the instructions from the journal.

Reviewer #3 (Remarks to the Author):

This study sheds new light on the complex relationship between maternal microbiota and type 2 innate immunity in infancy, revealing important cellular and molecular mechanisms for the causal relationship between maternal antibiotic exposure and increased risk of allergic inflammation in offspring. The design of this study is relatively novel and of profound clinical significance, but the reviewers still have doubts about the results of the study, which requires the author to revise and explain.

Major:

1. The ILC2 cells selected in this study were the main experimental objects, and the authors did not provide the most direct experimental evidence to prove the important role of ILC2 in this study. Moreover, the proportion of ILC2 in mouse lung tissue makes it difficult to extract primary cells, and it is necessary to prove the purity of the obtained cells and the possibility of differentiation.

The importance of ILC2 in the initiation and progression of allergic airway inflammation has been well documented (Licona-Limón P, Nat Immunol. 2013 PMID: 23685824). In this study, using ILC2-deficient mice (*Rora^{fl/fl}Il7r^{Cre}* mice) and adoptive transfer of ILC2s into immunodeficient NCG mice, we demonstrated that elevated neonatal ILC2 responses contributed to the aggravated airway inflammation

in offspring in response to maternal antibiotic exposure. We improved the clarity of the manuscript during revision. The purity of ILC2s from lung was verified by flow cytometric analysis (higher than 97%), which was provided in supplemented Gating strategies for this paper. To obtain enough number of ILC2s for experiments, IL-33 challenge model is usually used to expand *in vivo*.

A

ILC2s were sorted from neonatal lung ($CD45^+Lin^-CD90.2^+ST2^+CD25^+$), the purity was confirmed by flow cytometric analysis.

2. It needs to be confirmed whether the NCG mice selected by the authors will cause errors in experimental results due to metabolic disorders.

We understand the reviewer's concern. Due to the cross-talk between immune system with metabolism, such as ILC2s in white adipose tissue may limit the development of obesity (Brestoff, *et al.* Nature. 2015. PMID: 25533952). NCG mice is widely used in the field of ILC2 study (including our recent publication Cao, *et al.* Immunity, 2023. PMID: 36693372), which provide a model to investigate the intrinsic regulation of ILC2 function. Because NCG mice were used as recipients of ILC2 adoptive transfer for both control and experimental groups, we did not compare the NCG mice with other recipients, so we do not think the metabolic status of NCG mice affect the results of this study.

3. The authors describe in detail the changes in fetal rats associated with symptoms as they grow older, but do not provide any explanation of the mechanism. In the whole study, logical sorting and direct evidence are a little weak.

We understand the reviewer's concern. In this study, we showed that the offspring mice born to maternal antibiotics exposure displayed higher susceptibility of allergic airway

inflammation, in which enhanced ILC2 responses in neonates contributed to this phenotype. Mechanistic studies showed that maternal antibiotic exposure reduced the production of butyrate from breast milk, which downregulated IFN1 signaling in ILC2 in pups via feeding. The downregulation of IFN1 signaling enhanced ILC2 responses. Perinatal supplementation of butyrate counteracted the effects of maternal antibiotic exposure on allergic inflammation in offspring. These observations provide an early opportunity for allergic inflammation. We improved the logic flow by revising the statement as necessary.

4. Fig 7: Whether the authors considered the effect of butyrate treatment on pregnant mice without antibiotic treatment. How to confirm the concentration, use time and dosage of butyrate. What is the key effect of butyrate on pregnant mice? What is the metabolism and distribution of butyrate in pregnant mice?

Microbiota-derived SCFAs exert broad immunomodulatory functions to shape the immune system, which play a beneficial role in the hosts. Butyrate could drive colonic Treg differentiation and maintain gut homeostasis (Furusawa et al. Nature 2013, PMID: 24226770). After absorption into enterocytes, butyrate enters the circulation and regulate immune responses in the peripheral tissues. Butyrate could bind with its receptors expressed on target immune cells or *via* the inhibition of histone deacetylases (HDAC), elicits intracellular signaling and modulate the function of immune cells (Dalile, *et al.* Nat Rev Gastroenterol Hepatol. 2019. PMID: 31123355). Butyrate produced by maternal microbiota during pregnancy could secret into breastmilk and pass to the neonates during feeding. Butyrate in human breast milk was considered to be protective biomarker for food allergy (Paparó et al. Allergy. PMID: 33043467). In addition to immune system, maternal SCFAs also play an important role in the health of offspring, such as neuroplasticity, cognitive and social functions (Radford-Smith et al. Proc Natl Acad Sci U S A. 2022. PMID: 35197280; Liu et al. Cell Metab. 2021 PMID: 33651981). In this study, we aimed to investigate the protective role of maternal butyrate in the prevention of allergic airway inflammation in offspring upon antibiotic

exposure during pregnancy. Considering that SCFAs could be produced by microbiota under steady state condition, we did not study the effect of butyrate treatment on pregnant mice without antibiotic treatment, when butyrate is sufficient.

In order to avoid the adverse effects of antibiotic exposure to early development of fetus and avoid fetus loss, antibiotic exposure was administered to pregnant mice from embryonic day 10 to day 14. The dosage we used was based on the literature (Yu et al. Pharmacol Res. 2020. PMID: 32679183).

We provided the explanation in Discussion (page 18, 2nd paragraph).

Minor

1. The pulmonary edema and the changes of respiratory function in mice can show the injury of lung tissue more comprehensively.

Thanks for the reviewer's suggestions. Since we do not have machines for pulmonary edema and the changes of respiratory function available, we used eosinophilic infiltration, type 2 cytokine concentration in lung homogenates, as well as histological staining were used to evaluate the inflammation in lungs (Our recent publications: Cao et al. Immunity, 2023, PMID: 36693372; Liu et al. J Exp Med., 2022, PMID: 35044462).

2. The authors can select markers of ILC-2 cells for immunofluorescence or immunohistochemical staining to show its distribution in lung tissue.

We performed ILC2 immunofluorescence staining of ILC2s in lungs in previous publication (Liu et al. J Exp Med., 2022, PMID: 35044462).

REVIEWERS' COMMENTS

Reviewer #1 (Remarks to the Author):

The authors have effectively addressed all the concerns raised, resulting in a significantly improved manuscript. The revisions have strengthened the overall quality of the work.

Reviewer #2 (Remarks to the Author):

In their revised manuscript XU et al. still claimed the impact of maternal microbiota on mouse lung ILC2 functions through a mechanism that mobilized type I IFN pathway and bacteria derived butyrate.

The authors have thoughtfully addressed my primary concern regarding the role of maternal versus newborn microbiota by incorporating an analysis of the pups' microbiota (Fig. S3e and f). These findings demonstrate that both maternal and pups' microbiota undergo modifications, that can lead to alterations in butyrate production in both instances.

On various occasions, the authors judiciously emphasized that maternal antibiotic treatment is responsible for the observed effect, which is indeed accurate. A scientific paper's claims should align with the evidence presented. However, the current manuscript does not distinguish between the influence of maternal and offspring microbiota.

As a result, it is no longer possible to claim:

- in the title "Maternal microbiota inhibits ILC2 activation in neonates via induction of IFN1 signaling "

- in the abstract: "These observations demonstrate an essential role for the maternal microbiota in the control of type 2 innate immunity at the neonatal stage, which provides a therapeutic window for treating asthma in early life. "

-page 4: « These observations provide insights into the importance of the maternal microbiota in the suppression of type 2 innate immunity in early life. "

-page 9 : "These observations indicate that IFN- β , via its receptor IFNAR1, may mediate the effect of the maternal microbiota on ILC2 in neonates. "

The authors should stick to claim on page 19: "It uncovers an important cellular and molecular mechanism underlying the causal relationship between maternal antibiotic

exposure and the increased risk of allergic inflammation in offspring. "

The answers to the remaining questions are acceptable.

Reviewer #3 (Remarks to the Author):

During this revision, the authors addressed the reviewer's concerns in a point-to-point manner. Building upon the original manuscript, the authors maintained the novelty of this study while supplementing it with more rigorous experiments to substantiate the conclusions, without altering the overall structure and findings. We believe that this study now meets the criteria for publication.

Reviewer #1

The authors have effectively addressed all the concerns raised, resulting in a significantly improved manuscript. The revisions have strengthened the overall quality of the work.

We thank the reviewer's positive comments.

Reviewer #2

In their revised manuscript XU et al. still claimed the impact of maternal microbiota on mouse lung ILC2 functions through a mechanism that mobilized type I IFN pathway and bacteria derived butyrate.

The authors have thoughtfully addressed my primary concern regarding the role of maternal versus newborn microbiota by incorporating an analysis of the pups' microbiota (Fig. S3e and f). These findings demonstrate that both maternal and pups' microbiota undergo modifications, that can lead to alterations in butyrate production in both instances.

On various occasions, the authors judiciously emphasized that maternal antibiotic treatment is responsible for the observed effect, which is indeed accurate. A scientific paper's claims should align with the evidence presented. However, the current manuscript does not distinguish between the influence of maternal and offspring microbiota.

As a result, it is no longer possible to claim:

- in the title "Maternal microbiota inhibits ILC2 activation in neonates via induction of IFN1 signaling "

The title was changed to "Maternal antibiotic exposure enhances ILC2 activation in neonates via downregulation of IFN1 signaling"

- in the abstract: "These observations demonstrate an essential role for the maternal microbiota in the control of type 2 innate immunity at the neonatal stage, which provides a therapeutic window for treating asthma in early life. "

It was changed to "These observations demonstrate an essential role for the microbiota in the control of type 2 innate immunity at the neonatal stage, which provides a

therapeutic window for treating asthma in early life.

-page 4: « These observations provide insights into the importance of the maternal microbiota in the suppression of type 2 innate immunity in early life. "

It was changed to “These observations provide insights into the importance of the microbiota in the suppression of type 2 innate immunity in early life. "

-page 9 : “These observations indicate that IFN- β , via its receptor IFNAR1, may mediate the effect of the maternal microbiota on ILC2 in neonates. "

It was changed to “These observations indicate that IFN- β , via its receptor IFNAR1, may mediate the effect of the microbiota on ILC2 in neonates. "

The authors should stick to claim on page 19: “It uncovers an important cellular and molecular mechanism underlying the causal relationship between maternal antibiotic exposure and the increased risk of allergic inflammation in offspring. "

We made the corresponding changes and do not emphasize maternal microbiota in the revised manuscript, considering maternal antibiotic exposure affected both maternal and neonatal microbiota.

The answers to the remaining questions are acceptable.

We thank the reviewer’s valuable suggestions.

Reviewer #3

During this revision, the authors addressed the reviewer's concerns in a point-to-point manner. Building upon the original manuscript, the authors maintained the novelty of this study while supplementing it with more rigorous experiments to substantiate the conclusions, without altering the overall structure and findings. We believe that this study now meets the criteria for publication.

We thank the reviewer’s positive comments.